



# Volatile organic compounds and ozone air pollution in an oil production
# region in northern China
Tianshu Chen[1], Likun Xue[1,2*], Penggang Zheng[1], Yingnan Zhang[1], Yuhong Liu[1], Jingjing Sun[1], Guangxuan
Han[3], Hongyong Li[1], Xin Zhang[1,4], Yunfeng Li[1,4], Hong Li[4], Can Dong[1], Fei Xu[1,2], Qingzhu Zhang[1],
Wenxing Wang[1]
[1]Environment Research Institute, Shandong University, Ji'nan, Shandong, China.
[2]Shenzhen Research Institute of Shandong University, Shenzhen, Guangdong, China.
[3]Key Laboratory of Coastal Environmental Process and Ecology Remediation, Yantai Institute of Coastal Zone Research, Chinese
Academy of Sciences, Yantai, Shandong, China.
[4]Chinese Research Academy of Environmental Sciences, Beijing, China.
*Correspondence to*: Likun Xue (xuelikun@sdu.edu.cn)
**Abstract.**
Oil and natural gas (O&NG) exploration presents a significant source of atmospheric volatile organic
compounds (VOCs), which are central players of tropospheric chemistry and contribute to formations of
ozone ($O_3$) and secondary organic aerosols. The impacts of O&NG extraction on regional air quality have
been investigated in recent years in North America, but have long been overlooked in China. To assess the
impacts of O&NG exploration on tropospheric $O_3$ and regional air quality in China, intensive field
observations were conducted during February-March and June-July 2017 in the Yellow River Delta, an oil
extraction region in northern China. Very high concentrations of ambient VOCs were observed at a rural
site, with the highest alkane mixing ratios reaching 2498 ppbv. High $O_3$ episodes were not encountered
during wintertime but were frequently observed in summer. The emission profiles of VOCs from the oil
fields were directly measured for the first time in China. The chemical budgets of $RO_x$ radicals and $O_3$ were
dissected with a detailed chemical box model constrained by in-situ observations. The highly abundant
VOCs facilitated strong atmospheric oxidizing capacity and $O_3$ formation in the region. Oxygenated VOCs
(OVOCs) played an essential role in the $RO_x$ production, OH loss, and radical recycling. Photolysis of
OVOCs, $O_3$ and HONO, as well as ozonolysis reactions of unsaturated VOCs were major primary sources of
$RO_x$. $NO_x$ was the limiting factor of radical recycling and $O_3$ formation. This study underlines the important
impacts of O&NG extraction on atmospheric chemistry and regional air quality in China.



## 1. Introduction

Oil and natural gas (O&NG) compose the most significant fraction of global energy consumption and play an essential role in the industry, economy and social development. By the end of 2017, O&NG consumption accounted for approximately 58% of global primary energy consumption (The British Petroleum Company plc, 2018). In recent years, with the breakthroughs in exploration and extraction technologies for tight oil and shale gas such as horizontal drilling and hydraulic fracturing (EIA, 2014), the unconventional O&NG production has experienced explosive growth in the United States, resulting in an upward trend of O&NG production since 1980s (EIA, 2018). Increases in O&NG production are also projected in other countries with abundant reservoirs of shale oil/gas in the near future (EIA, 2014). O&NG production emits a large amount of air pollutants to the atmosphere, causing different levels of air pollution problems in the O&NG extraction region and its surrounding areas (Schnell et al., 2009; Edwards et al., 2013). The growth in O&NG production has indeed raised increasing concerns on the deteriorated air quality, public health, and climate in North America (Alvarez et al., 2012; McKenzie et al., 2012; Adgate et al., 2014; Colborn et al., 2014; Field et al., 2014).

Potential air pollutant emission sources during the O&NG production include deliberate venting and flaring, fugitive emissions, diesel engines for power supply, and leakage from infrastructure and transport (Adgate et al., 2014). Such activities have been shown to result in the increase of volatile organic compounds (VOCs) and nitrogen oxides (NOx) in the ambient air (Allen et al., 2013; Helmig et al., 2014; Warneke et al., 2014). Photochemical oxidation of VOCs in the presence of NOx produces ozone ($O_3$), a secondary pollutant with adverse effects on human health, vegetation, materials, and climate (National Research Council, 1992). Several field campaigns have observed unusually high levels of wintertime $O_3$ in oil and gas field basins in U.S., including Uintah Basin (Edwards et al., 2013; Edwards et al., 2014; Lee et al., 2014) and Upper Green River Basin (Schnell et al., 2009; Carter and Seinfeld, 2012). Such high wintertime $O_3$ episodes occur under the combined action of specific meteorological conditions and chemical processes. The favourable meteorological conditions include a shallow boundary layer, calm winds, and increased photolysis flux induced by the snow deposition (Schnell et al., 2009; Carter and Seinfeld, 2012; Ahmadov et al., 2015). In terms of atmospheric chemistry processes, the accumulated high concentrations of VOCs lead to a significant increase in $O_3$ production efficiency, and radicals generated by photolysis of oxygenated VOCs (OVOCs) also play an important role (Edwards et al., 2013; Edwards et al., 2014). In addition, the O&NG production also affects $O_3$ formation and air quality during other seasons, especially in





summer. Rodriguez et al. (2009) used a regional chemical transport model (CAMx) to assess the impacts of
O&NG operation on $O_3$ pollution in the western U.S., and found the enhancement in the maximum daily 8-h
average $O_3$ (MDA8 $O_3$) by considering O&NG emissions can reach up to 9.6 ppbv in southwestern
Colorado and north-western New Mexico. Using the same model, Kemball-Cook et al. (2010) indicated that
emissions from Haynesville Shale can explain up to 5 ppbv of MDA8 $O_3$ enhancement within Northeast
Texas and Northwest Louisiana. Other works also found that the O&NG extraction activities pose important
effects on regional $O_3$ levels in summertime (Olaguer, 2012; Rutter et al., 2015; Vinciguerra et al., 2015;
McDuffie et al., 2016).

9       The O&NG exploration activities are very active in China, with crude oil and natural gas production

both ranking the sixth in the world (EIA, 2017; Statista, 2018). China is also rich in shale resources, with the
reserves of shale gas and shale oil ranking the first and third in the world, respectively (EIA, 2014). It is
expected that China's future O&NG exploration will further increase and may pose increasingly important
effects on the atmospheric environmental issues. Currently, $O_3$ pollution has become a major air quality
concern in China, with monitored $O_3$ concentrations exceeding the national ambient air quality standard
frequently in the metropolitan areas nationwide (Xue et al., 2014a; Wang et al., 2017). Available long-term
observations also demonstrated significant upward trends in surface $O_3$ levels in the last two decades over
China (Ding et al., 2008; Wang et al., 2009; Xu X. et al., 2008; Xue et al., 2014b; Sun et al., 2016; Xu W. et
al., 2018). A large number of studies have dedicated to understand the formation mechanisms of $O_3$
pollution and identified the major sources of $O_3$ precursors (particularly VOCs) in China (e.g., Zhang et al.,
2008; Yuan et al., 2012; Dang et al., 2015; Shao et al., 2016; Zhao et al., 2016; Wang et al., 2017).
However, O&NG extraction has long been overlooked as an important source of VOCs, compared to the
other anthropogenic activities such as industry, power plants, transportation, biomass burning, etc. To the
best of our knowledge, to date there is no report that has assessed the impacts of O&NG exploration on
VOCs and $O_3$ pollution in China.
To fill this gap, two intensive measurement campaigns were conducted at a rural site surrounded by
open oil fields in the Yellow River Delta (YelRD) region, an important oil extraction area in China, during
February–March and June–July of 2017. A large suite of parameters including $O_3$, CO, NO, $NO_2$, $NO_y$, $SO_2$,
HONO, $C_1$-$C_{10}$ hydrocarbons, $C_1$-$C_8$ carbonyls, aerosol properties, and meteorological parameters were
measured in-situ. Air samples were also collected from oil wells to characterize the source profiles of VOCs
in the oil field. A detailed chemical box model was then constrained with the above-mentioned in-situ




observations to dissect the chemistry of $O_3$ formation, atmospheric oxidative capacity, and radical budgets.
Overall, this study provides some new insights into the emission characteristics of VOCs from oil fields and
their effects on the atmospheric oxidation processes and regional $O_3$ pollution in China.

## 2. Materials and Methods

### 2.1. Site description

We target the YelRD region for assessing the impacts of oil field emissions on the VOC and $O_3$
pollution. The YelRD is located to the south of Bohai Sea and in the northern part of Shandong Province. It
includes Dongying, Binzhou and parts of Weifang, Dezhou, Zibo and Yantai cities, with a total area of
26,500 square kilometers and a population of 9.85 million (Figure 1). It is abundant in natural resources and
hosts the third largest oilfield in China (i.e., Shengli Oilfield). Active O&NG exploration has made it one of
China's largest petrochemical industry bases. In addition, the YelRD estuary is a typical estuarine wetland
ecosystem and is rich in ecological resources. Furthermore, it is located at the junction of the Beijing-
Tianjin-Hebei region and Shandong Peninsula, the most polluted regions in North China, with distances of
approximately 300, 200 and 190 km away from Beijing, Tianjin and Ji'nan, respectively. Therefore, it may
also suffer from regional transport of aged continental air masses from these metropolitan areas under the
influence of winter monsoons.
Two phases of field campaigns were carried out in winter-spring (from February 9 to April 1) and
summer (from June 1 to July 10) 2017 at the YelRD Ecological Research Station of Coastal Wetland
(37.75°N, 118.97°E; 1 m above sea level), Chinese Academy of Sciences. This site lies roughly 32 km to
the northeast of Dongying urban area and 10 km to the west of the Bohai Sea (Figure 1). It is a typical rural
site surrounded by open oil fields and without any other anthropogenic emission sources nearby. There are
two intensive oil production areas near the site. One is mainly distributed in the coastal area (about 10 km to
the northeast), while the other is in the urban area (about 30 km to the southwest). In view of the regional
scale, the observation site is constrained by both aged continental air masses transported from the Beijing-
Tianjin-Hebei region and clean marine air from the Bohai Sea, making it an excellent platform to study the
interaction between anthropogenic pollution and the natural background air in the North China Plain (NCP).
All in-situ measurement instruments were housed in a temperature-controlled container, and the sampling
inlets were mounted on top of the container with an altitude of about 5 m above the ground. Source samples



from the nearby oil and gas wells were also collected to obtain the source profiles of VOCs from the oil
field. Details of the sampling site can be found elsewhere (Zhang et al., 2019).
**2.2. Measurement techniques**
A large suite of chemical species and meteorological parameters were measured. Briefly, $O_3$ was
monitored by an ultraviolet photometric analyzer (*Thermo Environmental Instruments (TEI) Model 49C*).
NO and $NO_y$ were measured by a chemiluminescence instrument (*Advanced Pollution Instrumentation*
*(API) Model T200U*) equipped with an externally placed molybdenum oxide catalytic converter. $NO_2$ was
observed with a Cavity Attenuated Phase Shift (CAPS) analyzer that is highly selective for true $NO_2$ (*API,*
*Model T500U*). $SO_2$ was observed using a pulsed ultraviolet fluorescence analyzer (*TEI, Model 43C*). CO
was detected using a gas filter correlation non-dispersive infrared analyzer (*API Model 300U*). The particle
number size distributions between 5 nm and 350 nm were measured by a Wide-Range Particle Spectrometer
(*WPS, Model 1000XP, MSP Corporation, USA*), while those in the range of 300 nm to 10 μm were
monitored by a Handheld Particle Counter (*Model 9306, TSI, USA*). $PM_{2.5}$ mass concentrations were
monitored using a SHARP analyzer (*Thermo Scientific Model 5030*). HONO was detected by a long path
absorption photometer named LOPAP (*QUMA GmbH, Germany*). Meteorological parameters including
wind direction, wind speed, temperature, and relative humidity (RH) were continuously observed by a
weather station (*PC-3, Jinzhou Sunshine*). Photolysis frequencies of $H_2O_2$, HCHO, HONO, $O_3$, $NO_3$, and
$NO_2$ were observed by a CCD-detector spectrometer (*Metcon GmbH, Germany*). The time resolution was 1-
min averaged for trace gases and photolysis frequency, 5-min averaged for meteorological parameters, and
30-min averaged for $PM_{2.5}$.
Whole air samples were collected with clean and evacuated 2-L stainless steel canisters for
quantification of methane and $C_2$-$C_{10}$ non-methane hydrocarbons (NMHCs). The samples were mainly
collected on sunny days (with a small part on cloudy days) during selected pollution episodes, with each
sample taken every 2~3 h for 30 seconds from 7:00 to 19:00 local time (LT) in June-July and from 6:00 to
21:00 LT in February-March. In addition, 7 samples were taken at 00:00 LT during the winter-spring
campaign. The purpose of such VOC sampling strategy is to better recognize the VOC pollution
characteristics in this area and to facilitate detailed modelling analysis of $O_3$ pollution events. Whole air
samples were also collected exactly in the surroundings of oil wells and petrochemical industrial areas using
the same method. A total of 111 ambient samples (including 58 samples in winter-spring and 53 samples in
summer 2017) as well as 21 source samples (including 18 oilfield samples and 3 petrochemical plant



samples) were taken in this study. After sampling, concentrations of methane and $C_2$-$C_{10}$ NMHCs were then
quantified by gas chromatography (GC) separation followed by flame ionization detection (FID), mass
spectrometry detection (MSD) and electron capture detection (ECD) at the laboratory of the University of
California at Irvine (Simpson et al., 2010; Xue et al., 2013). The detection limit is 0.01 ppmv for methane
and 3 pptv for $C_2$-$C_{10}$ NMHCs (Simpson et al., 2010).
Carbonyl samples were collected by adsorption of ambient air in a 2,4-dinitrophenylhydrazinecoated
sorbent cartridge (*Waters Sep-Pak DNPH–silica*) at a flow rate of 0.5 L min$^{-1}$. An $O_3$ scrubber is attached to
the front of the cartridge to avoid $O_3$ interference. The sampling strategy is similar to that of VOC canister
samples. Specifically, the carbonyl samples were taken during selected episodes every 3 h from 6:00 to
21:00 LT in winter-spring and every 2 h from 7:00 to 19:00 LT in summer (the sampling time for each
sample in winter-spring and summer was 3 h and 2 h, respectively). A total of 128 ambient samples
(including 58 samples in winter-spring and 70 samples in summer) and 10 source samples were taken at the
rural site and in the oil fields, respectively. After the campaign, the samples were analyzed with the high-
performance liquid chromatography (HPLC) for quantification of 14 $C_1$-$C_8$ carbonyl species (Yang et al.,

15    2018).

**2.3. Observation-Based Chemical Box Model**
The Observation-Based Model for investigating the Atmospheric Oxidative Capacity and
Photochemistry (OBM-AOCP) was used to simulate the in-situ atmospheric photochemical processes and to
quantify the $O_3$ production rate, OH reactivity and radical budgets ($RO_x$: OH, $HO_2$ and $RO_2$). This model has
been successfully adopted in many previous studies (e.g., Xue et al., 2014a; Xue et al., 2016; Yang et al.,
2018; Li et al., 2018; Sun et al., 2018). In short, it is based on the latest version of the Master Chemical
Mechanism (MCM v3.3.1), a nearly explicit mechanism describing the gas phase chemical reactions that
involve 143 primary VOC species (Saunders et al., 2003). In addition to the existing reactions in MCM
v3.3.1, OBM-AOCP also incorporates over 200 reactions which represent the oxidation of VOCs by
chlorine radical (Xue et al., 2015) and heterogeneous processes involving reactive nitrogen oxides (Xue et
al., 2014a). Physical processes such as dry deposition and dilution mixing in the boundary layer are also
taken into account, and details can be found elsewhere (Xue et al., 2014a).
OBM-AOCP is able to simultaneously quantify the $O_3$ production rate, atmospheric oxidizing capacity
(AOC), OH reactivity, as well as the primary production, recycling and termination rates of $RO_x$ radicals. It





tracks and calculates the individual reaction rate of almost all the reactions in the MCM, including the free
radical chemistry. Among them, the sum of oxidation rates of various pollutants (CO, VOCs, $NO_x$, $SO_2$,
etc.) by the major oxidants (i.e., OH, $O_3$, $NO_3$ and Cl) is regarded as the AOC (Xue et al., 2016). The
reaction rates of OH with CO, VOCs, $NO_x$, $SO_2$, HONO, $HNO_3$, and $HO_2NO_2$ are computed as the OH
reactivity. Primary sources of OH, $HO_2$ and $RO_2$ include the photolysis reactions of $O_3$, HONO,
formaldehyde and other OVOCs as well as reactions of VOCs with $O_3$ and $NO_3$ radicals (Xue et al., 2016).
Related reactions were grouped into a dozen major routes of production, recycling and loss for quantifying
the $RO_x$ chemical budget (Xue et al., 2016). The $O_3$ production rate is calculated from the difference
between the oxidation rates of NO by $HO_2$ and $RO_2$ radicals and the loss rates of $O_3$ and $NO_2$ (Xue et al.,
2014a). Details of the above chemistry calculation can be found elsewhere (Xue et al., 2014a; Xue et al.,

11 2016).

Measured data of $O_3$, $SO_2$, CO, NO, $NO_2$, HONO, $J(NO_2)$, temperature, and RH were averaged to a
time resolution of 5 minutes to constrain the model. Besides, measured concentrations of $CH_4$, $C_2$-$C_{10}$
NMHCs, and $C_1$-$C_8$ carbonyl compounds were interpolated to a time resolution of 30 minutes for model
inputs. For the nighttime data, when direct observations were generally unavailable, $CH_4$ and $C_2$-$C_{10}$
NMHCs (except isoprene) concentrations were interpolated according to their linear regressions with CO,
and concentrations of isoprene were interpolated based on the linear relationship with temperature (Yang et
al., 2018). The nighttime OVOC data were interpolated according to the multiple linear regressions with CO
and $O_3$ (Yang et al., 2018). Such approximation was mainly to facilitate the pre-run of the model, and should
not affect the formal daytime modelling results. Photolysis frequencies within the model were adjusted by
the solar zenith angle and the measured $J(NO_2)$ (Saunders et al., 2003). The model starts at 00:00 LT and
pre-runs for 4 days under constraints of input data to stabilize the species which were not measured in the
field campaign, and the daytime modelling results of the last day were subject to further analyses.
**3. Overview of $O_3$ and VOC pollution**
The overall air quality and meteorological conditions measured during the two-phase campaign are
presented in Figure 2. Descriptive statistics of major trace gases, aerosols, and meteorological parameters
are summarized in Table 1. Seasonal variability of air pollution and weather is clearly illustrated. The winter
and early spring (i.e., February-March) is featured by cold weather and higher levels of primary air
pollutants. All the trace gases (except for $O_3$) and $PM_{2.5}$ showed significantly higher concentrations in
February and March than in summer (June-July). This can be explained by the shallow boundary layer, less



active photochemistry, and additional emissions from residential heating in winter-spring. In contrast, $O_3$ exhibited much higher levels in summer, mainly corresponding to the more intense photochemical formation as a result of the hot weather and strong solar radiation. Elevated $O_3$ concentrations were frequently observed during the summer campaign, with 22 non-attainment days (defined as the day when the maximum hourly $O_3$ concentration exceeds China's National Ambient Air Quality Standard, Grade II, 93 ppbv) throughout the 40-day measurement period. The maximum hourly $O_3$ value was recorded at 177 ppbv in summer. These observations demonstrate the severity of photochemical air pollution in the YelRD region.

$O_3$ pollution was also encountered in early spring. In March, two $O_3$ non-attainment days were identified with a maximum hourly $O_3$ mixing ratio of 106 ppbv. When looking at the MDA8 $O_3$, the number of non-attainment days (with MDA8 $O_3$ exceeding 75 ppbv) increased to five in March 2017. However, no $O_3$ episodes occurred in February. This is quite different from the recent observations in U.S. that have found very high levels of $O_3$ in winter in the oil basin (Schnell et al., 2009; Edwards et al., 2014). We examined the observed chemical environments and weather conditions in the YelRD region. As detailed below, there were abundant $O_3$ precursors, especially VOCs, in this study region, which would sustain as much as photochemical $O_3$ formation. The major difference between this study and the U.S. efforts lies in the weather conditions. As proposed by Ahmadov et al. (2015), snow cover is a prerequisite for the occurrence of wintertime $O_3$ episodes in the U.S. oil basins. During the wintertime observation period, the weather was quite dry and only small amounts of snowfall occurred during the nighttime of February 21. The snow cover was very thin and it quickly disappeared with increase of temperature under the influence of a subsequent high-pressure system. Furthermore, the YelRD region is usually affected by strong winds in winter (Fig. 2) due to its flat and coastal topography. Thus, the meteorological conditions encountered in the present study were unfavourable for the occurrence of winter $O_3$ episodes. Similarly, $O_3$ episodes were also not observed in the Uintah basin in the snow-free winter of 2012 (Edwards et al., 2014). More observations are still needed to examine the wintertime $O_3$ issues in the oil extraction areas of China.

Table 2 documents the statistics of individual VOC species observed in the present study. Obviously, the ambient air in the YelRD region is very rich in VOCs, in particular alkanes which accounted for the majority (i.e., 84.3% for winter-spring and 70.6% for summer) of the measured NMHCs. Extremely high levels of VOCs were frequently observed at the study site, although it is located in a remote coastal area. The maximum concentrations of total NMHCs were 2823 ppbv and 176 ppbv in winter-spring and summer, respectively. These samples were heavily affected by the gas leakage from the surrounding oil fields and



will be discussed further in Section 4. Besides, elevated concentrations of light olefins such as ethene,
propene, and butenes were also detected, especially during the winter and early spring when the
photochemical oxidation was less active. This was mainly attributed to the emissions from refining industry
in the YelRD region, which is well known as an important base for petrochemical industry in north China. A
number of refining plants are indeed located to the southwest and north of the sampling site. Such VOC-rich
atmosphere is expected to efficiently facilitate $O_3$ production with a certain amount of $NO_x$. Furthermore,
similar to other primary pollutants, all of the VOC compounds (except for cyclopentane and isoprene)
showed a typical seasonal variation with higher concentrations in winter-spring and lower levels in summer.
Figures 3-4 present the average diurnal variation patterns of major trace gases (including VOCs), $PM_{2.5}$,
and meteorological parameters during the two campaigns. All the pollutants showed well-defined diurnal
profiles which can be explained by the evolution of planetary boundary layer, local emissions, and
atmospheric photochemistry. Specifically, $O_3$ showed a broad afternoon concentration peak with a trough in
the early morning in both seasons. The other primary pollutants (e.g., CO, $SO_2$ and $NO_x$) and $PM_{2.5}$
exhibited higher concentrations in the morning and the lowest levels in the afternoon. VOCs generally
showed higher levels during the nighttime or the early morning and lower mixing ratios during the day, with
long-chain alkenes (comprising isoprene, 3-methyl-1-butene, 2-methyl-1-butene, alpha-pinene, and beta-
pinene) as an exception that shows an opposite diurnal pattern (Fig. 4). A noteworthy result is the fast
accumulation of $O_3$ during the morning period. For example, the average increases in $O_3$ concentrations in
the morning (i.e., 06:00–12:00 LT) were 49.2 ppbv and 30.2 ppbv in summer and winter-spring,
respectively. Considering the remote nature of the study site, such rapid $O_3$ increase suggests the strong in-
situ photochemical formation in this VOC-rich area. This will be further quantified with the model in
Section 6.

### 4. Emission profiles of VOCs from oil fields

To characterize the VOC emissions from the oil fields in China, 18 whole air samples were taken
exactly close to the oil extraction machines in the open oil fields. The data can provide direct insights into
the composition profile of VOCs from Chinese oil field emissions. Regional background of VOC species
was calculated as the average of the lowest 10[th] percentile of measurement data at the study site, and was
subtracted from the oilfield source data to derive the VOC emission profiles. Figure 5 shows the measured
oilfield emission profiles of VOCs in the YelRD region. It is obvious that oilfield emissions are dominated
by alkanes. On a concentration basis, light alkanes ($C_2$-$C_5$), long-chain alkanes ($C_6$-$C_{10}$), alkenes, and





aromatics account for 83.7%, 8.7%, 3.1%, and 2.9% of the total measured NMHCs, respectively. The top
ten abundant species (in proportion) are propane (25.3%), ethane (22.1%), *n*-butane (13.6%), *i*-butane
(8.3%), *i*-pentane (7.8%), *n*-pentane (6.0%), ethene (1.9%), *n*-hexane (1.8%), ethyne (1.6%), and *2*-
Methylpentane (1.3%).
Note that all the aforementioned calculations are based on the median VOC emission profile shown in
Figure 5. Since alkanes are major components of crude oil and natural gas, measured oilfield emissions in
this study are believed to be due to the leakage of oil and natural gas in this oilfield region. To our
knowledge, this should be the first piece of direct measurements of oilfield VOC emission profiles in China,
which is valuable for better understanding the emissions of O&NG production and can be used for future air
quality modelling studies.
Figure 6 compares the oilfield emission profile in the YelRD region with those obtained from
measurements adjoin to or surrounded by the U.S. oil fields. Overall, the measured VOC speciation patterns
agree well with each other, although the absolute VOC concentrations vary case by case. For example, the
VOC concentrations in the oilfield in this study are generally higher than or comparable to those in the Fort
Worth Basin, Denver-Julesburg Basin, and Upper Green River Basin, but are much lower than those
measured in the Uintah Basin during the wintertime $O_3$ episodes. Such differences should be mainly caused
by different atmospheric dilution conditions during the sampling campaigns. The extremely high VOC
levels in the Uintah Basin can be ascribed to the strong inversion under unfavourable weather conditions
(Neemann et al., 2015). There are also some differences in the detailed VOC speciation between the YelRD
oilfield emissions and those in U.S. The fraction of $C_2$-$C_5$ light alkanes in the YelRD oil fields was lower
than those in the Uintah Basin (93.9%), Fort Worth Basin (90.4%), and Denver-Julesburg Basin (92.9%)
(ERG, 2011; Gilman et al., 2013; Koss et al., 2015). In comparison, the loadings of long-chain alkanes
(8.7%) and aromatics (2.9%) were higher in the YelRD oilfield than in the U.S. oil basins (4.2-6.9% for
long-chain alkanes, <1.6% for aromatics). Such VOC speciation was attributed to the fact that oil extraction,
rather than natural gas production, dominates in this study area.
As mentioned above, the ambient air at the sampling site may be influenced by the oilfield emissions
significantly. To verify this issue, all the ambient VOC data were subject to the Tukey Test (Seo, 2006), and
11 samples were identified as 'abnormal sample'. According to the VOC concentrations and speciation, the
ambient VOC samples can be classified into 3 categories. Type 1 contains four 'abnormal samples' and
these samples have the highest concentrations for most species, especially alkanes, butenes, and aromatics




(Fig. 6). Type 2 includes seven 'abnormal samples' which have almost the same chemical speciation and
absolute concentrations (only with slightly lower levels of light alkanes) as the oilfield emission profiles
(Fig. 6). The remaining 100 'normal samples' are classified as Type 3. Compared with the oilfield emission
profile, they have similar chemical speciation but lower concentrations. In terms of the sampling time, Types
1 and 2 samples were mainly collected in the early morning or at midnight, whilst most of the Type 3
samples were taken during the daytime. Figure 7 shows the scatter plots of *i*-pentane versus *n*-pentane for
the three identified ambient VOC types as well as the oilfield source data. Because *i*-pentane is generally
recognized as tracer of gasoline, the ratio of *i*-pentane/*n*-pentane can be adopted to diagnose the potential
impact of O&NG operations on the VOC measurements in the O&NG extraction region (Gilman et al.,
2013). As shown in Figure 7, Type 2 (1.2) and Type 3 (1.3) samples have comparable *i*-pentane/*n*-pentane
ratios to the oilfield source data (1.0). Meanwhile, Type 1 samples have a much higher ratio of 4.5, which is
similar to the signature of gasoline emissions (4.87) (Lu et al., 2003). In view of the above analyses, we
propose that Type 1 samples were affected by short-term leakage from the surrounding refinery and oil
storage areas; Type 2 samples were heavily influenced by the O&NG extraction activities in the oil fields;
and the 'normal' Type 3 samples were also affected by the O&NG extraction in this region. This indicates
that the VOC-rich environment in the YelRD region is mainly influenced by the O&NG extraction activities.
**5. Atmospheric oxidative capacity and radical chemistry**
In the following sections, we examine the detailed photochemical processes that occurred during the $O_3$
pollution episodes. As few episodes were encountered during winter-spring, here we focus on the
summertime $O_3$ pollution events. Nine severe $O_3$ episodes (i.e., 8, 9, 14, 15, 16, 18, 29, 30 June, and 9 July
2017) with the maximum hourly $O_3$ concentrations exceeding 100 ppbv and with concurrent comprehensive
observation data were sorted out for chemical box modelling analyses. Detailed chemical budgets of $RO_x$
radicals and $O_3$ were quantified by the OBM-AOCP. Simulation results for different cases were generally
similar. Below we present the results that have been averaged across all selected episodes.
Figure 8 shows the average diurnal variations of OH and $HO_2$ during the $O_3$ episode days. High levels
of $HO_x$ radicals were simulated by the model. The daily maxima of OH and $HO_2$ concentrations were 4.7-
7.0 $\times 10^6$ molecules $cm^{-3}$ and 10.3-14.1 $\times 10^8$ molecules $cm^{-3}$, with mean values of $5.9\times 10^6$ molecules $cm^{-3}$
and $12.5\times 10^8$ molecules $cm^{-3}$, respectively. Model-predicted concentrations of $HO_x$ radicals in the rural area
of YelRD are higher than those at Heshan (a rural site in the Pearl River Delta, southern China) and Mace
Head (a coastal site in Ireland) (Smith et al., 2006; Tan et al., 2018). Comparable noontime maxima $HO_x$





concentrations were observed in some polluted urban areas, such as Tokyo and Houston (Kanaya et al.,
2007; Mao et al., 2010). This demonstrates the strong potential of atmospheric oxidation in the YelRD
region. A noteworthy result is the OH concentration peak occurring in the morning (at around 10:00 LT),
which is different from the most common results showing noontime OH peaks with intense solar radiation
(Rohrer and Berresheim, 2006). To a large extent, the diurnal pattern of OH follows that of NO (see Fig. 3),
suggesting the important role of NO in OH chemistry at the sampling site. Considering the VOC-rich
condition and relatively low levels of $NO_x$ (e.g., observed average concentrations of NO are 0.43 and 0.23
ppb during 9:00-12:00 and 12:00-16:00 LT, respectively), efficient radical propagation of $OH \rightarrow RO_2 \rightarrow HO_2$
is expected and the abundance of NO should be the limiting factor in the recycling of $HO_2$ to OH. The
higher ratios of $HO_2/OH$ (~257) in this study also indicate that the $HO_2+NO \rightarrow NO_2+HO$ reaction is the rate-
determining step of the radical recycling. Similar phenomenon was also found at Backgarden (a VOC-
saturated and $NO_x$-limited environment) in the PRD region (Lu et al., 2012).
The strong atmospheric oxidizing capacity (AOC; defined as the oxidation rates of all reduced
substances by major oxidants) was confirmed by the model calculation, and is shown in Figure 9. The daily
maxima and daily mean values of AOC during the selected episodes were in the range of $0.7\text{-}1.8 \times 10^8$ and
$2.6\text{-}4.8 \times 10^7$ molecules $cm^{-3}$ $s^{-1}$, respectively. AOC levels in the YelRD region are comparable to those
obtained in some urban areas (Elshorbany et al., 2009; Xue et al., 2016), but are higher than those derived
from rural areas (Geyer et al., 2001; Li et al., 2018). As expected, OH is the predominant oxidant during the
daytime, accounting for 85.3±16.4% of AOC. $NO_3$ is the major oxidant at nighttime (18:00-6:00 LT),
contributing 46.8±17.1% of AOC, followed by $O_3$ (27.0±7.9%) and OH (26.2±17.8%). Figure 10 elucidates
the 24-hour evolution and partitioning of the chemical loss of OH radical (also known as the OH reactivity
or $K_{OH}$). $K_{OH}$ in this study (23.3±5.6 $s^{-1}$) is significantly higher than those determined from some rural sites
such as Hok Tsui (9.2±3.7 $s^{-1}$) (Li et al., 2018), Nashville (11.3±4.8 $s^{-1}$) (Martinez et al., 2003), and
Whiteface Mountain (5.6 $s^{-1}$) (Ren et al., 2006a), and is comparable to those obtained in some polluted urban
and suburban areas (Ren et al., 2006b; Whalley et al., 2016). OVOCs (including the oxidation intermediates
and products of VOCs in the model) were the dominant contributor (69.1±7.2%) to $K_{OH}$. CO, $NO_x$, alkenes,
alkanes, and aromatics are the other important reactants, explaining 13.2±2.5%, 5.6±4.1%, 4.4±1.5%,
3.6±1.2%, and 1.6±0.5% of $K_{OH}$, respectively. The relatively higher fraction of alkanes is probably due to
the highly abundant alkanes in the YelRD region as a result of influences from the oilfield emissions.





Figure 11 presents major primary sources of OH, $HO_2$ and $RO_2$ radicals quantified in the YelRD region,
and the detailed $RO_x$ radical budget is summarized in Figure 12. Photolysis of OVOCs is identified as the
dominant radical source, with daytime (6:00-18:00 LT) average production rates of 2.15±1.40 ppbv $h^{-1}$ for
$HO_2$ (of which 1.10±0.79 ppbv $h^{-1}$ is from formaldehyde alone) and 0.86±0.53 ppbv $h^{-1}$ for $RO_2$,
respectively. $O_3$ photolysis is the second largest source of $RO_x$ and the predominant primary source of OH
(1.22±1.10 ppbv $h^{-1}$). HONO photolysis is the third largest source and supplies OH at an average rate of
0.49±0.48 ppb $h^{-1}$ during the daytime. The contribution of HONO photolysis is higher than that of $O_3$
photolysis in the early morning (e.g., before 9:00 LT), but then becomes significantly lower with the
decrease in HONO concentrations and photochemical formation of $O_3$. Note that the model was constrained
by the observed HONO data. Ozonolysis reactions of unsaturated VOCs are also important radical sources,
accounting for 0.26±0.11, 0.17±0.07 and 0.14±0.07 ppbv $h^{-1}$ of OH, $HO_2$ and $RO_2$, respectively, on a
daytime average basis. In comparison, $NO_3$+VOCs reactions are only a minor radical source (for $RO_2$ only).
The above analysis illustrates the significant role of OVOCs (both primary carbonyls and secondary
compounds formed from oxidation of abundant VOCs) in the primary production of radicals and thus
initiation of atmospheric oxidation processes. The dominance of photolysis of OVOCs in the atmospheric
photochemistry was also found during the wintertime $O_3$ episodes in the Uintah basin (Edwards et al., 2014).
As shown in Figure 12, the radical recycling processes were generally efficient and approximately 4-6
times faster than the primary radical production. This is ascribed to the high abundances of VOCs in the
study region, despite the restriction from the relatively low $NO_x$ concentrations. In terms of radical
termination, the cross reactions of radicals such as $HO_2$+$HO_2$ and $HO_2$+$RO_2$ were the most important
processes with daytime average contributions of 0.55±0.48 and 1.12±0.94 ppbv $h^{-1}$, respectively. In
comparison, the reactions of $RO_x$ with $NO_x$ (i.e., OH+$NO_2$ and $RO_2$+NO) contributed 1.19±1.62 ppbv $h^{-1}$ to
the radical sink. Such results are not surprising given the VOC-rich and low-$NO_x$ chemical environment at
our study site. Overall, the radical budget analysis elucidates the strong atmospheric oxidizing capacity, the
importance of OVOCs, and the limiting role of $NO_x$ in the VOCs-rich atmosphere of the YelRD region.
**6. Ozone formation mechanism**
We also examined the ozone formation mechanisms for the summertime episode days. Figure 13 shows
the average $O_3$ production, destruction (including dry deposition), and net rates during the nine cases. Strong
photochemical formation of $O_3$ was clearly illustrated, with daily maximum net $O_3$ production rates of 14.5-
38.7 ppbv $h^{-1}$ and daytime-average rates (6:00-18:00 LT) of 9.8-19.6 ppbv $h^{-1}$, respectively. The $O_3$



production intensity in the rural area of the YelRD is higher than that derived from a rural site downwind of Beijing (Changping), and comparable to those in polluted suburban areas downwind of Shanghai and Lanzhou (Xue et al., 2014a). Interestingly, the $O_3$ production rate shows its maxima in the morning period (at around 10:00 LT) followed by a significant decrease in the afternoon, which differs from general results from previous studies showing noontime or afternoon peaks. This pattern is similar to that of OH and NO (Figs. 3 and 8), and should be due to the lower concentrations of NO in the afternoon. In the VOCs-rich YelRD region, a certain amount of NO in the morning is enough to sustain efficient $O_3$ production. In the afternoon, $NO_x$ has been photochemically consumed due to its short lifetime and thus becomes the limiting factor in $O_3$ formation (note that $O_3$ production rate is defined as the reaction rates of $HO_2+NO$ and $RO_2+NO$). This also explains the observed unusual diurnal variation of $O_3$ (Fig. 3), with significant increase during the morning period and constant or reduced levels in the afternoon.

The relationships between $O_3$ and its precursors were further diagnosed by the relative incremental reactivity (RIR) calculation using the OBM-AOCP model. RIR is defined as the ratio of the change in $O_3$ production rate to changes in precursor concentrations, and it can be used as an indicator for assessing the effect of precursor reduction on $O_3$ formation (Cardelino and Chameides, 1995). A number of sensitivities modelling runs were conducted for individual episode days with 20% reduction in the input concentrations of each target $O_3$ precursor group. As presented in Figure 14, simulation results for most cases are similar. $O_3$ production was most sensitive to $NO_x$ concentrations, as indicated by the highest positive RIR values. This is expected as the aforementioned analyses suggest the limiting role of $NO_x$ in radical recycling and $O_3$ production. Alkenes, especially long-chain alkenes, showed moderate positive RIR values, indicating they controlled $O_3$ formation to some extent as well. Alkanes and aromatics are usually in high abundances owing to the extensive oil extraction in the YelRD region, showing minor RIR values and were not the limiting factors for $O_3$ formation. Overall, reducing $NO_x$ emissions would be the most effective strategy for mitigating photochemical air pollution in the YelRD region.

**7. Conclusions**

We combined intensive field observations with chemical box modelling to understand the characteristics of VOC emissions from oil fields and their impacts on atmospheric chemistry and $O_3$ pollution in the YelRD region, North China. Influenced by the O&NG extraction and petrochemical industry, this area is featured by a VOCs-rich atmosphere with extremely high levels of alkanes. $O_3$ pollution episodes occurred frequently in summertime. Meanwhile, no events were encountered in winter-



spring because of the unfavourable weather conditions for $O_3$ formation. The VOC chemical speciation from
the oil field emissions was detected for the first time in China in this study. Driven by the high abundances
of VOCs on a regional scale, strong atmospheric oxidizing capacity and intense $O_3$ formation were
confirmed by observation-based modelling analyses. OVOCs played a dominant role in OH reactivity and
hence radical recycling, and were the major source of $RO_x$ radicals. Photolysis of $O_3$ and HONO were also
found to be important radical sources. The radical termination processes were governed by radical cross
reactions under the high-VOCs and low-$NO_x$ conditions. RIR analysis indicated that $O_3$ formation was
mainly in a $NO_x$-controlled regime, and reducing $NO_x$ emissions would be an effective way to control $O_3$
pollution in the YelRD region. In summary, this study emphasized the key role of O&NG extraction in the
photochemical air pollution and regional atmospheric chemistry in the oil extraction regions of China, and
the results are helpful for formulating the anti-pollution strategies in the YelRD and other similar oil-
extracting regions.
**Data availability.**
The data that support the results are available from the corresponding author upon request.
**Author contributions.**
LX designed the study. TC, PZ, YL, JS and HYL conducted the field campaigns. GH provided logistics
for the field campaigns. HL, XZ and YL analyzed the OVOC samples. TC analysed the measurement data.
TC and YZ conducted the chemical box modelling analyses. TC and LX wrote the paper. GH, DC, HL, FX,
QZ and WW revised the manuscript.
**Competing interests.**
The authors declare that they have no conflict of interest.
**Acknowledgments.**
The authors thank Mr. Changli Yang, Rui Li, and Xinfeng Wang for their help in the field study, and
thank Ms. Zeyuan Li and Xue Yang for their efforts in data analysis and discussion. We thank Prof. Donald
Blake from the University of California at Irvine for the laboratory analyses of VOC samples, and appreciate
the University of Leeds for provision of the MCM v3.3.1. This study is funded by the National Natural
Science Foundation of China (grant No.: 41675118), Shandong Provincial Science Fund for Distinguished





Young Scholars (ZR2019JQ09), Shenzhen Science and Technology Research and Development Funds
Grant (JCYJ20160510165106371), the Qilu Youth Talent Programme of Shandong University, the Jiangsu
Collaborative Innovation Center for Climate Change, and the Taishan Scholars (ts201712003).

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





1    **Table 1.** Descriptive statistics of hourly concentrations of trace gases, $PM_{2.5}$ and meteorological parameters
2    at the rural site in the YelRD region

| Species/Parameter | February-March, 2017 | | | June-July, 2017 | | |
|---|---|---|---|---|---|---|
| | Mean±SD | Median | Max | Mean±SD | Median | Max |
| $O_3$ (ppbv) | 34±17 | 36 | 106 | 65±28 | 60 | 177 |
| CO (ppbv) | 530±331 | 463 | 2667 | 428±221 | 373 | 1728 |
| NO (ppbv) | 1.39±3.11 | 0.17 | 46.92 | 0.31±0.50 | 0.17 | 19.03 |
| $NO_2$ (ppbv) | 10.08±8.84 | 7.50 | 55.34 | 3.47±3.34 | 2.36 | 37.32 |
| $NO_y$ (ppbv) | 19.84±16.40 | 16.85 | 86.74 | 10.13±7.74 | 8.44 | 69.58 |
| $SO_2$ (ppbv) | 4.68±5.16 | 2.99 | 44.68 | 2.10±2.71 | 1.14 | 34.09 |
| $PM_{2.5}$ (μg/m$^3$) | 66.7±56.4 | 50.6 | 247 | 41.1±33.1 | 31.5 | 167.9 |
| TEMP (℃) | 5.8±4.8 | 5.8 | 18.6 | 25.9±4.5 | 26.1 | 36.8 |
| RH (%) | 69±18 | 73 | 100 | 76±16 | 82 | 99 |





1 **Table 2.** Descriptive statistics of measured VOC concentrations at the rural site in the YelRD region

| Compound | February-March, 2017 | | | June-July, 2017 | | |
|---|---|---|---|---|---|---|
| | Mean±SD | Median | Max | Mean±SD | Median | Max |
| $CH_4$ | 2116±159 | 2084 | 2869 | 223±282 | 2184 | 3704 |
| Ethane | 7.094±4.143 | 6.091 | 21.986 | 5.092±5.211 | 3.394 | 29.878 |
| Propane | 29.640±88.873 | 5.380 | 470.670 | 5.740±7.448 | 3.353 | 38.081 |
| i-Butane | 24.456±87.067 | 1.581 | 484.988 | 1.983±2.500 | 1.155 | 14.660 |
| n-Butane | 38.417±134.908 | 2.546 | 732.394 | 3.399±4.579 | 2.231 | 25.996 |
| i-Pentane | 30.687±110.933 | 1.361 | 585.862 | 1.693±2.334 | 1.042 | 11.956 |
| n-Pentane | 7.209±22.698 | 0.909 | 123.655 | 1.213±1.782 | 0.781 | 10.122 |
| n-Hexane | 0.255±0.464 | 0.093 | 2.337 | 0.324±0.510 | 0.097 | 2.362 |
| n-Heptane | 1.041±2.296 | 0.315 | 11.441 | 0.394±0.575 | 0.249 | 3.313 |
| n-Octane | 0.518±1.376 | 0.105 | 7.126 | 0.121±0.177 | 0.071 | 0.951 |
| n-Nonane | 0.167±0.383 | 0.052 | 2.058 | 0.052±0.054 | 0.036 | 0.314 |
| n-Decane | 0.251±0.747 | 0.046 | 3.586 | 0.042±0.029 | 0.035 | 0.168 |
| 2,3-Dimethylbutane | 0.097±0.220 | 0.035 | 1.052 | 0.027±0.018 | 0.021 | 0.092 |
| 2-Methylpentane | 0.153±0.295 | 0.052 | 1.529 | 0.069±0.102 | 0.033 | 0.513 |
| 3-Methylpentane | 0.646±1.366 | 0.195 | 7.062 | 0.256±0.476 | 0.121 | 2.769 |
| 2,4-Dimethylpentane | 0.489±0.964 | 0.179 | 4.411 | 0.183±0.300 | 0.090 | 1.717 |
| Cyclopentane | 0.187±0.476 | 0.020 | 1.990 | 0.028±0.031 | 0.014 | 0.134 |
| Methylcyclopentane | 1.329±2.931 | 0.361 | 12.773 | 0.369±0.402 | 0.232 | 1.698 |
| Cyclohexane | 2.081±6.728 | 0.133 | 32.069 | 0.136±0.213 | 0.067 | 1.112 |
| Methylcyclohexane | 0.441±1.143 | 0.090 | 5.376 | 0.129±0.191 | 0.056 | 0.920 |
| Ethene | 2.203±1.311 | 2.013 | 5.925 | 1.076±1.047 | 0.709 | 4.662 |
| Propene | 1.362±2.283 | 0.588 | 14.442 | 0.624±1.344 | 0.163 | 8.805 |
| 1-Butene | 0.203±0.376 | 0.069 | 1.711 | 0.085±0.244 | 0.025 | 1.627 |
| i-Butene | 0.878±1.428 | 0.254 | 4.472 | 0.055±0.074 | 0.034 | 0.491 |
| trans-2-Butene | 0.110±0.130 | 0.042 | 0.461 | 0.027±0.036 | 0.011 | 0.107 |
| cis-2-Butene | 0.094±0.099 | 0.054 | 0.360 | 0.050±0.058 | 0.028 | 0.135 |
| 1,3-Butadiene | 0.084±0.092 | 0.047 | 0.324 | 1.317±3.880 | 0.022 | 11.664 |
| Isoprene | 0.112±0.219 | 0.032 | 0.929 | 2.738±1.701 | 2.497 | 7.113 |
| 3-Methyl-1-butene | 0.036±0.034 | 0.022 | 0.120 | 0.024±0.019 | 0.016 | 0.061 |
| 2-Methyl-1-butene | 0.046±0.052 | 0.029 | 0.255 | 0.025±0.026 | 0.012 | 0.071 |
| alpha-Pinene | 0.424±1.345 | 0.021 | 5.700 | 0.015±0.005 | 0.014 | 0.028 |
| beta-Pinene | 0.122±0.125 | 0.026 | 0.291 | 0.020±0.012 | 0.015 | 0.037 |
| Ethyne | 3.055±1.964 | 2.868 | 13.553 | 2.261±1.759 | 1.731 | 8.450 |
| Benzene | 1.073±0.567 | 1.064 | 2.537 | 0.709±0.533 | 0.539 | 2.852 |
| Toluene | 14.378±50.177 | 0.828 | 250.922 | 0.507±0.510 | 0.285 | 2.317 |
| Ethylbenzene | 0.648±1.781 | 0.157 | 9.058 | 0.107±0.099 | 0.078 | 0.632 |
| m/p-Xylene | 1.542±4.599 | 0.260 | 21.785 | 0.157±0.184 | 0.090 | 1.117 |
| o-Xylene | 0.573±1.662 | 0.104 | 7.440 | 0.072±0.075 | 0.047 | 0.465 |
| Styrene | 0.173±0.339 | 0.034 | 1.507 | 0.036±0.054 | 0.016 | 0.216 |
| i-Propylbenzene | 0.096±0.201 | 0.019 | 0.732 | 0.029±0.020 | 0.020 | 0.083 |
| n-Propylbenzene | 0.118±0.282 | 0.028 | 1.113 | 0.023±0.018 | 0.016 | 0.084 |
| m-Ethyltoluene | 0.269±0.757 | 0.042 | 3.538 | 0.035±0.046 | 0.019 | 0.232 |
| p-Ethyltoluene | 0.164±0.387 | 0.034 | 1.440 | 0.032±0.032 | 0.022 | 0.154 |
| o-ethyltoluene | 0.175±0.385 | 0.034 | 1.452 | 0.030±0.026 | 0.020 | 0.111 |
| 1,3,5-Trimethylbenzene | 0.197±0.423 | 0.030 | 1.447 | 0.031±0.025 | 0.023 | 0.081 |
| 1,2,4-Trimethylbenzene | 0.290±0.804 | 0.047 | 3.748 | 0.042±0.055 | 0.022 | 0.254 |
| 1,2,3-Trimethylbenzene | 0.108±0.180 | 0.038 | 0.743 | 0.032±0.025 | 0.019 | 0.099 |
| Total NMHC | 171.177±527.177 | 30.041 | 2823.177 | 29.706±30.278 | 23.189 | 175.661 |

2 Units: ppbv.





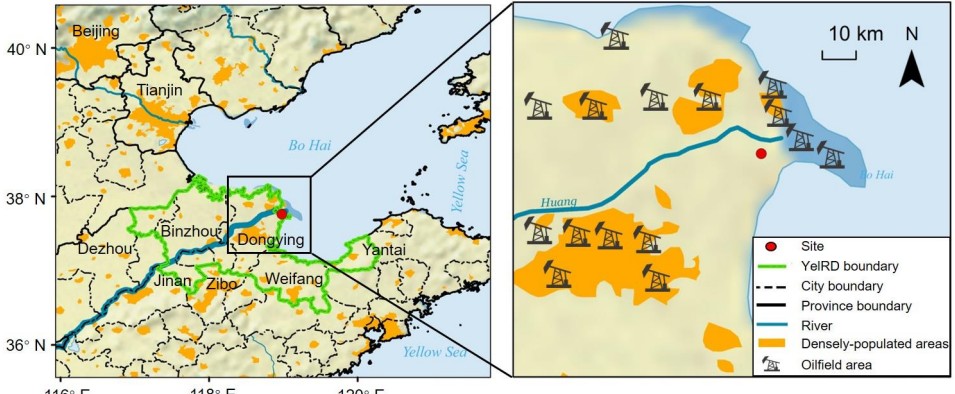

**Figure 1.** Map showing the location of the Yellow River Delta. The right map shows the surroundings of the sampling site and the approximate positions of the oilfield areas (Base map: made with Natural Earth).





**Figure 2.** Time series of trace gases, PM2.5, and meteorological parameters measured at the study site during February-March and June-July 2017.



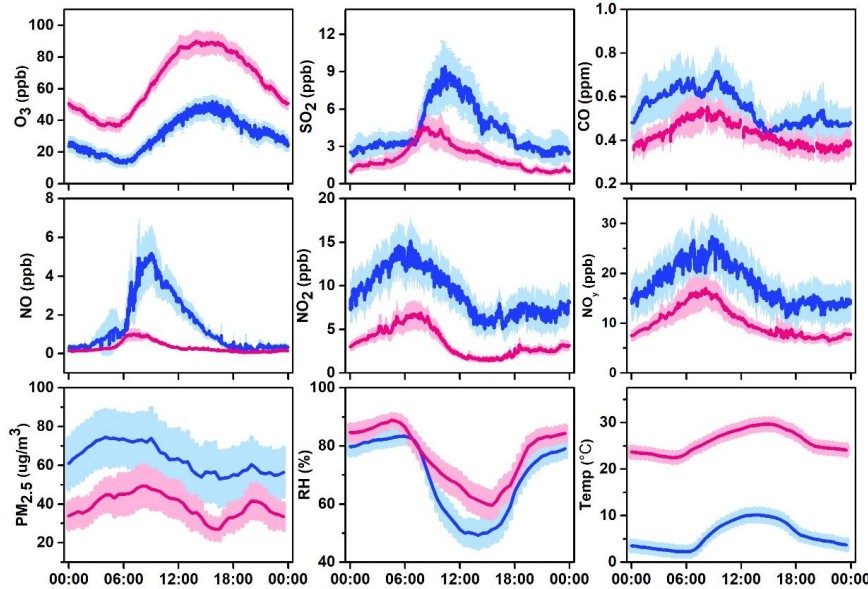

2 **Figure 3.** Average diurnal patterns of trace gases, PM2.5, and meteorological parameters at the study site
3 during February-March and June-July 2017. Error bars indicate half standard deviation of the mean (blue
4 line: February-March, red line: June-July).





**Figure 4.** Average diurnal variations of light alkanes, long-chain alkanes, light alkenes, long-chain alkenes, alkyne, BTEX and other aromatics at the study site (left column: February-March, right column: June-July).

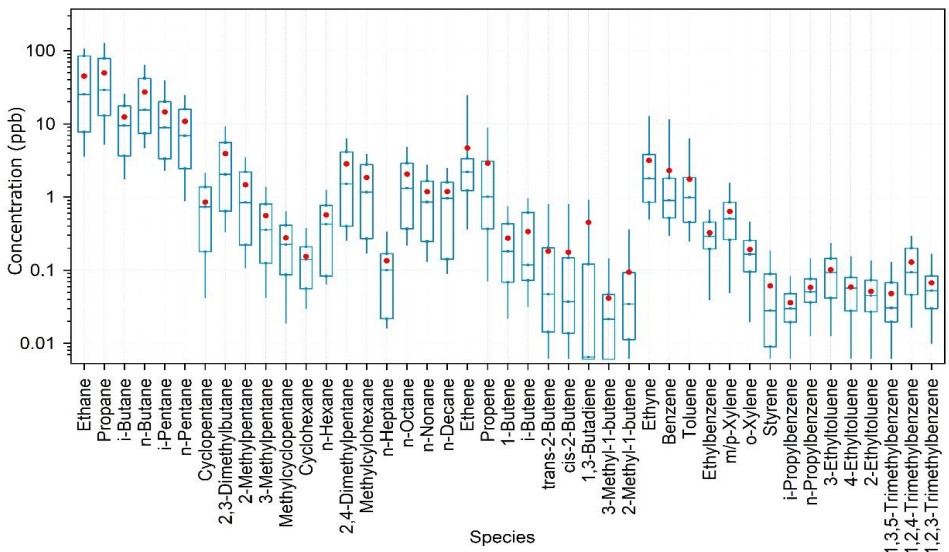

**Figure 5.** VOC source profile of the oil field emissions in the YelRD region. The box plot provides the 10th,
25th, 50th, 75th and 90th of the source sample data, and red dot gives the average of the data. Note that the
regional background has been subtracted from the source data.

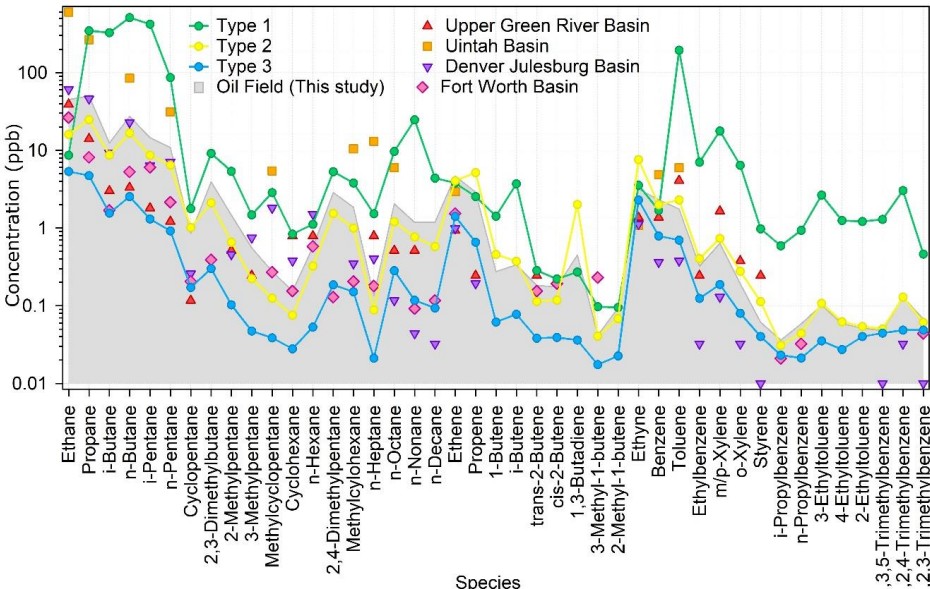

**Figure 6.** Comparison of the VOC composition of oil field samples (grey area) with three types of ambient
samples in this study and in four U.S. oil fields (ERG, 2011; Field et al., 2015; Gilman et al., 2013; Koss et
al., 2015).

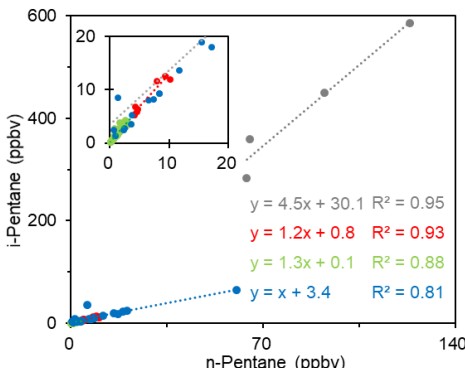

**Figure 7.** Scatter plot and regression lines of *i*-pentane versus *n*-pentane for the three types of ambient samples and oilfield samples (grey: Type 1, red: Type 2, green: Type 3, blue: Source; refer to the main text for the description of different types of data)

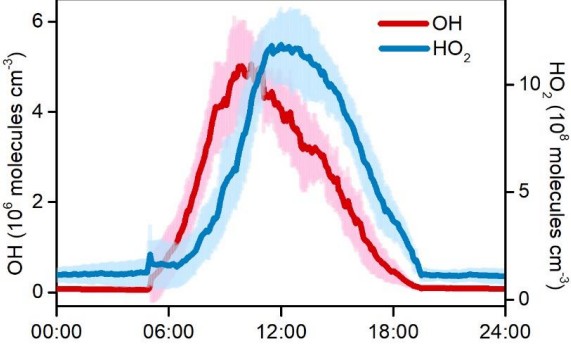

**Figure 8.** Simulated average diurnal variations of OH and $HO_2$ during the nine $O_3$ pollution episodes. The shaded areas indicate the standard deviations of the mean.

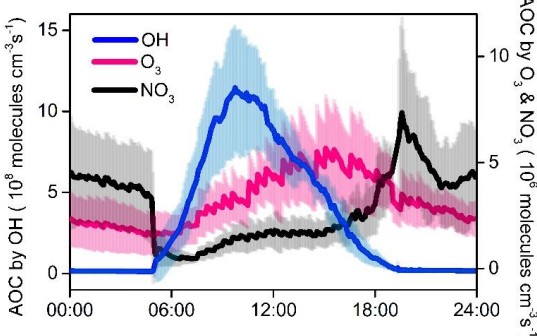

**Figure 9.** Model-calculated average oxidizing capacity of OH, $O_3$ and $NO_3$ during the summertime $O_3$ pollution episodes. The error bars indicate the standard deviations of the mean.





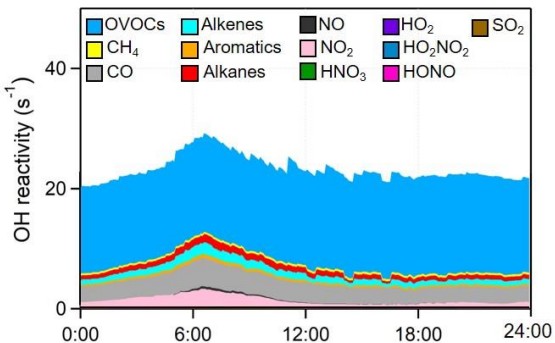

**Figure 10.** Model-calculated average OH reactivity ($K_{OH}$) and its breakdown to the major reactants during
the summertime $O_3$ pollution episodes.





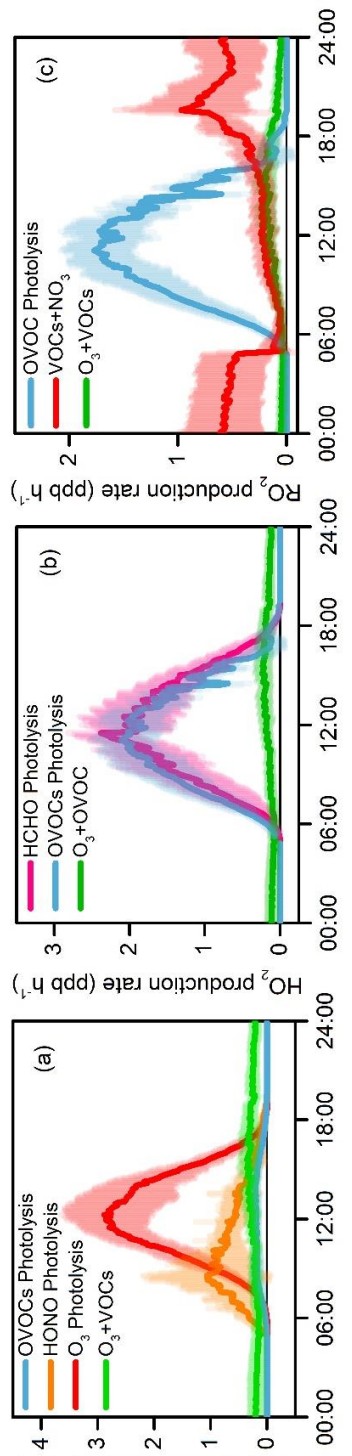

**Figure 11.** Simulated average primary production rates of (a) OH, (b) HO$_2$, and (c) RO$_2$ during the summertime O$_3$ pollution episodes. The error bars indicate the standard deviations of the mean.


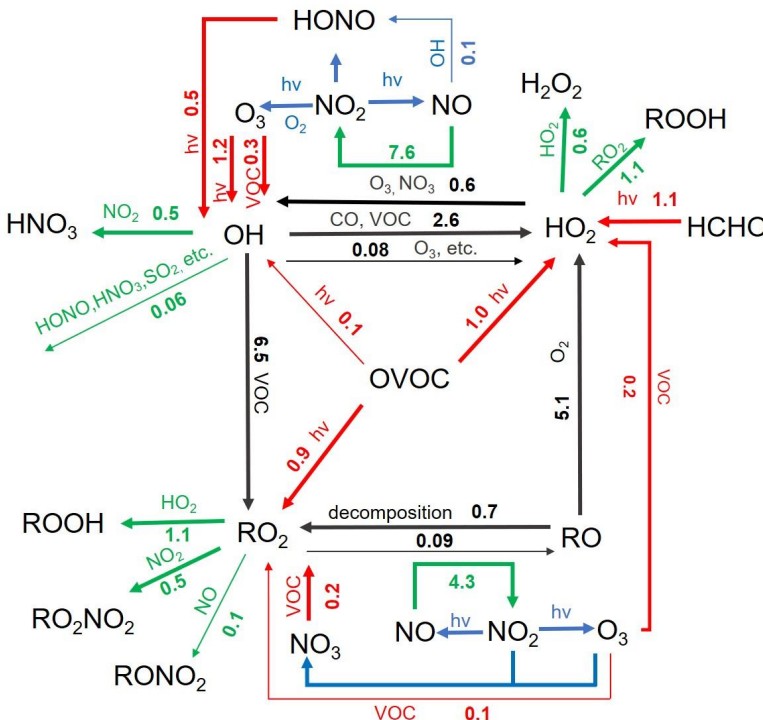

**Figure 12.** Daytime average ROx budget during the summertime $O_3$ episode days. The unit is ppb h$^{-1}$. The red, green and black lines indicate the production, destruction and recycling pathways of radicals, respectively.

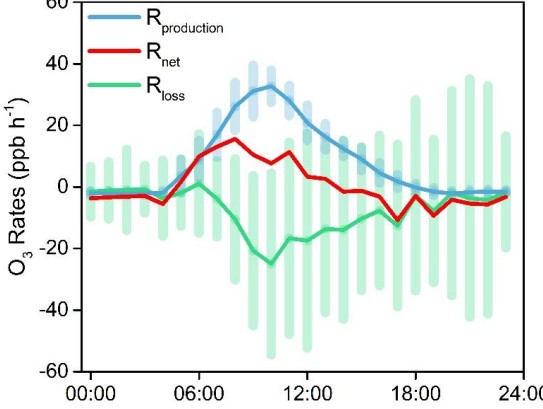

**Figure 13.** Simulated average $O_3$ production, destruction, and net rates during the summertime $O_3$ pollution episodes. The error bars indicated the standard deviations of the mean for $O_3$ production and destruction rates.

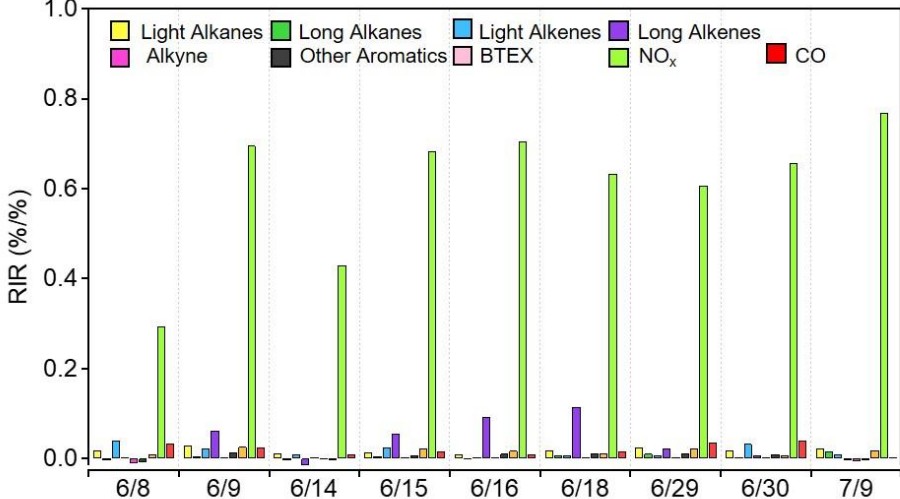

2  **Figure 14.** Model-calculated mid-day average (9:00-15:00 LT) RIRs for the major $O_3$ precursor groups
3  during the summertime $O_3$ episodes.