# Peer review of "Volatile organic compounds and ozone air pollution in an oil production region in northern China"

_Atmospheric Chemistry and Physics, 2019_

## Referee Comment (RC1) · Anonymous Referee #1 · 28 Dec 2019

General comments

This paper presents observations of NMHCs, OVOCs, O3, and other reactive trace gases made at a site in the Yellow River Delta during a winter-spring period and a summer period in 2017. The study area is very interesting because it is one of the largest oil and natural gas (O&NG) exploration areas in China. And it is a part of the North China Plain; a region suffers from severe air pollution. Some of the NMHCs and OVOCs samples were taken near oil wells and petrochemical industrial areas. The authors derived emission profiles of VOCs from oil fields. Such emission profiles were not available in China before. The concentrations of VOCs in ambient air over the

site were very high due to large emissions from O&NG exploration in the Yellow River Delta. The authors also studied the atmospheric oxidative capacity, radical budget, and ozone formation mechanism at the observation site using a box model constrained by measurements. The results show that O3 formation was mainly NOx-controlled due to high-VOCs and low-NOx conditions. OVOCs played a dominant role in OH reactivity and were the major source of ROx radicals. The low NOx level in summer limited the radical recycling.

The observational data reported in this paper are valuable. The emission profiles of VOCs from O&NG exploration are highly needed in air quality studies and emission control. The results from the box model study are interesting and important for such a special area. This paper is well written and within the scope of ACP. I recommend publication of this paper in ACP after the following minor points are appropriately addressed.

Specific comments and suggestions:

1. After reading this paper the readers would like to know the impacts of large VOCs emission in the Yellow River Delta on the air quality over the surrounding area. The summer concentration of NOx at the study site was low. However, the NOx levels over many other parts of the North China Plain are much higher. Photochemical O3 production may be very different when transported high-VOCs plume mixed with high-NOx air. To address this 3D simulation is needed, which may be out of the scope of this paper. But I think some discussions in this aspect are necessary.

2. Section 2.2 should include details about the calibrations and data quality.

3. Although HONO and OVOCs were observed, no observational data of these are presented. Since these species are very imported in your studies of atmospheric oxidative capacity and radical chemistry, I do not think these observational data can be omitted. I think HONO data should be included in Figures 2 and 3, and OVOCs data should be either presented in Figure 4 or in an extra figure.

[Figure]

4. P13, L1-4 and Figure 11: why is the "NO+HO2 = NO2 + OH" reaction not included as a source of OH? If you have any reason not to include this reaction in OH production, you should not state "Photolysis of OVOCs is identified as the dominant radical source" (L2-3) because your Figure 11a shows OVOCs photolysis has only a minor contribution to OH production.

5. Section 6, first paragraph and P9, L18-20: your measurements show a rapid morning increase of O3 concentration. This increase may not only be resulted from photochemistry but also from vertical mixing. I suggest that you integrate the net O3 production rate in Figure 13 to get a diurnal profile of O3 based on box model simulations and compare it with the observed diurnal profile shown in Figure 3.

6. Some of the box model results can be compared with those from Chinese megacities reported by Tan et al. (2019). [Tan et al., Daytime atmospheric oxidation capacity in four Chinese megacities during the photochemically polluted season: a case study based on box model simulation, Atmos. Chem. Phys., 19, 3493–3513, 2019.]

7. P3, L15-18: when it comes to long-term trends of surface ozone in China, "Ma et al., Significant increase of surface ozone at a rural site, north of eastern China, Atmos. Chem. Phys., 16, 3969-3977, 2016" is one of the few important papers and should be cited.

8. P4, L23-26: "In view of the regional scale, the observation site is constrained by both aged continental air masses transported from the Beijing-Tianjin-Hebei region and clean marine air from the Bohai Sea, making it an excellent platform to study the interaction between anthropogenic pollution and the natural background air in the North China Plain (NCP)". I do not think you can really sample "clean marine air from the Bohai Sea" or "natural background air" because you are so close to the oil wells.

9. P4, L27-P5, L2.: these sentences belong to section 2.2.

10. P5, L14: what is "SHARP"? This abbreviation should be explained.

11. P5, L21-30: did you use O3 scrubbers when taking NMHCs samples?

12. P7, L20-21: "Photolysis frequencies within the model were adjusted by the solar zenith angle and the measured J(NO2) (Saunders et al., 2003)." Why did you adjust photolysis frequencies within the model when you had the measured values?

13. P9, L14-17: this statement applies only to summer.

14. P9, L26-28: can you estimate errors in the so obtained emission profiles?

15. Figure 11b: Since you single out HCHO photolysis, "OVOCs Photolysis" here should be changed to "Photolysis of OVOCs other than HCHO" or similar.
* * *

---

## Referee Comment (RC2) · Anonymous Referee #2 · 16 Feb 2020

General Comment The impact of the O&NG exploration on the O3 formation in both summer and winter is one of the key issues in the studies of tropospheric chemistry. Negative impact of the O&NG exploration in North America had already been presented in a series of field studies in both United States and Canada including the warnings of serious wintertime ozone non-attainment for the Basin landscape. The study of O&NG exploration on the atmospheric chemistry is missing in China and has been nicely filled up by the current paper based on winter and summer field studies in Yellow River Delta (YelRD) which is a place famous for open oil fields in China. The dataset obtained in this study are very valuable and the data analysis is systematic and scientifically sound. The paper has been clearly written. Overall, I suggest to publish

the paper after the authors addressed the following comments.

Major Comment 1. It will be very valuable for the current paper to analyze the radical budget and the ozone production rates also for the winter campaign. And compare with the corresponding US studies to show from a chemical perspective why the ozone pollution is not appeared in the YeIRD region.

Specific Comment 1. Page 6, lines 1- 6, as described by the authors, the VOC samples were analyzed in the US lab which is far away. How did the authors make sure that the reactive compounds are not decayed away during the time span between sampling and the lab analysis?

2. The authors used "Atmospheric Oxidative Capacity" (page 6, line 17; page 11, line 17), "Atmospheric Oxidizing Capacity" (page 6, line 28; page 12, line 13), to denote AOC. I suggest the authors to use "Atmospheric Oxidation Capacity" for that. At least, it should be unified throughout the paper.

3. Page 7, lines 9-11, I don't think the authors can refer the calculations of the ozone production rates simply to previous papers. The detailed equation needs to be given here and the uncertainty of the calculations is worth to be analyzed.

4. Figure 2, Y title, 'jO1D' the number 1 shall be superscript.

5. Figure 4, from the plot of the subgroups of the NMHCs, it is not clear to the readers how the high concentrations up to hundreds and thousands of ppbv of NMHCs is make up?

6. Page 11, line 29 -30, why not compared your results to measurements of HOx radicals in NCP such as Tan et al., ACP, 2017. And Tan et al., ACP, 2018 needs updated (it is already published in ACP).

7. Page 12, line 21 – 29, there were a number of direct total OH reactivity measurements published for field studies in NCP and PRD by the PKU and FZJ groups. I suggest the authors shall compare those in addition to the comparison to US cities. To

my knowledge, the kOH in this study is within the range of the available measurement in China.

8. Figure 10, a lot of OVOCs is calculated in the model. The summed reactivity is much higher than that of their precursors (alkanes, alkenes, and aromatics) which is normally difficulty to achieve. The large accumulation of OVOCs in the model may be related with the lifetime of those OVOCs implemented in the model. Could the authors break-down the speciation of the calculated OVOCs and compare with the measurements as mentioned in the part of methods. The comparison between model and measurements for OVOCs could help the authors to define the lifetime of the OVOCs in the model.

9. Figure 11, the OH production rate from O3 photolysis seems to be a little large according to your O3 concentrations presented in Figure 3. I assume you calculated the photolysis rate of O3 (O3 + hv –> O1D), but the OH production needs a further reaction with H2O (O1D + H2O –> OH) which is competed by reaction with N2 and O2 (O1D+M–>O), the yield of OH is often around 10% of the photolysis rate of O3 depends on the H2O concentrations.

10. Figure 12, daytime average, please specify the exact time span of the hours

11. Figure 13, the O3 loss rate reached up to 20 ppb/h, what are the major reactions for that?

---

## Author Comment (AC1) · 25 Mar 2020

**Response to Reviewer 1:**

*General comments*

*This paper presents observations of NMHCs, OVOCs, $O_3$, and other reactive trace gases made at a site in the Yellow River Delta during a winter-spring period and a summer period in 2017. The study area is very interesting because it is one of the largest oil and natural gas (O&NG) exploration areas in China. And it is a part of the North China Plain; a region suffers from severe air pollution. Some of the NMHCs and OVOCs samples were taken near oil wells and petrochemical industrial areas. The authors derived emission profiles of VOCs from oil fields. Such emission profiles were not available in China before. The concentrations of VOCs in ambient air over the site were very high due to large emissions from O&NG exploration in the Yellow River Delta. The authors also studied the atmospheric oxidative capacity, radical budget, and ozone formation mechanism at the observation site using a box model constrained by measurements. The results show that $O_3$ formation was mainly NOx-controlled due to high-VOCs and low-NOx conditions. OVOCs played a dominant role in OH reactivity and were the major source of ROx radicals. The low NOx level in summer limited the radical recycling.*

*The observational data reported in this paper are valuable. The emission profiles of VOCs from O&NG exploration are highly needed in air quality studies and emission control. The results from the box model study are interesting and important for such a special area. This paper is well written and within the scope of ACP. I recommend publication of this paper in ACP after the following minor points are appropriately addressed.*

**Response:** we thank the reviewer for the positive comments and constructive suggestions to improve our manuscript. We have carefully considered all the review comments and revised the manuscript accordingly. Below we provide the original reviewer's comments *in black italic*, with our responses and changes in the manuscript in blue and red, respectively.

*Specific comments and suggestions:*

*1. After reading this paper the readers would like to know the impacts of large VOCs emission in the Yellow River Delta on the air quality over the surrounding area. The summer concentration of NOx at the study site was low. However, the NOx levels over many other parts of the North China Plain are much higher. Photochemical $O_3$ production may be very different when transported high-VOCs plume mixed with high-NOx air. To address this 3D simulation is needed, which may be out of the scope of this paper. But I think some discussions in this aspect are necessary.*

**Response:** thanks for the suggestion. We agree with the reviewer that it is important to assess the impacts of oilfield VOC emissions on the regional air quality in the surrounding areas, and

it is indeed an initial objective of this project. As the reviewer indicated, 3-D model simulations and emission inventory of oilfield emissions are needed to address this issue, which is out of the scope of this paper. A separate work that aims at developing an oilfield emission inventory of VOCs and assessing its impacts on regional ozone pollution by WRF-Chem model is now underway. The following discussions have been added in the revised manuscript to address this aspect.

"Nonetheless, the oilfield emissions of VOCs may have high potential to affect the regional air quality in the polluted YelRD and even the surrounding NCP regions, where the ambient $NO_x$ are usually abundant. The oilfield emitted VOCs may significantly contribute to the formations of $O_3$ and secondary organic aerosol on a regional scale. To address this issue, an oilfield emission inventory of VOCs and $NO_x$ as well as 3-dimensional chemical transport model simulations are needed. So far, the oilfield emission has not been included by the emission inventories in China. More efforts are urgently needed to develop accurate oilfield emission inventory and evaluating their impacts on the regional air quality and climate."

*2. Section 2.2 should include details about the calibrations and data quality.*

**Response:** the following descriptions have been provided in the revised manuscript.

"These trace gas analyzers were calibrated manually every three days during the measurement campaigns, including zero and span checks as well as conversion efficiency calibration of the MoO catalytic converter, with additional zero calibration automatically done every four hours for the CO instrument."

"All of the above measurement techniques have been successfully applied in many previous studies, and the detailed measurement principles, detection limits, quality assurance and quality control procedures can be found elsewhere (Xue et al., 2016; Simpson et al., 2016; Yang et al., 2018; Li et al., 2018)."

*3. Although HONO and OVOCs were observed, no observational data of these are presented. Since these species are very imported in your studies of atmospheric oxidative capacity and radical chemistry, I do not think these observational data can be omitted. I think HONO data should be included in Figures 2 and 3, and OVOCs data should be either presented in Figure 4 or in an extra figure.*

**Response:** in the revised manuscript, the HONO data has been included in Figures 2 and 3, and the OVOC data has been presented in Figure 4. The revised figures are as follows.

[Figure]

**Revised Figure 2.** Time series of major trace gases, PM$_{2.5}$, and meteorological parameters measured at the study site during February-March and June-July 2017.

[Figure]

**Revised Figure 3.** Average diurnal patterns of major trace gases, PM$_{2.5}$, and meteorological parameters at the study site during February-March and June-July 2017. Error bars indicate the half standard deviation of the mean (blue line: February-March, red line: June-July).

[Figure]

**Revised Figure 4.** Average diurnal variations of light alkanes, long-chain alkanes, light alkenes, long-chain alkenes, alkyne, BTEX, other aromatics, formaldehyde, and $C_2$-$C_8$ carbonyls at the study site (left column: February-March, right column: June-July).

*4. P13, L1-4 and Figure 11: why is the "NO+HO$_2$ = NO$_2$+OH" reaction not included as a source of OH? If you have any reason not to include this reaction in OH production, you should not state "Photolysis of OVOCs is identified as the dominant radical source" (L2-3) because your Figure 11a shows OVOCs photolysis has only a minor contribution to OH production.*

**Response:** here we only focus on the PRIMARY RO$_x$ radical sources, and the "NO+HO$_2$ = NO$_2$+OH" reaction was treated as a radical recycling process in the present study. The contribution of this recycling process to the OH production was also evaluated and presented in Figure 12. For clarity, the original statements have been modified as follows in the revised manuscript.

"Figure 11 presents major primary sources of OH, HO$_2$ and RO$_2$ radicals quantified in the YelRD region, and the detailed RO$_x$ radical budget is summarized in Figure 12. Photolysis of OVOCs is identified as the dominant primary RO$_x$ radical source, with daytime (6:00-18:00 LT) average production rates of 2.15±1.40 ppbv h$^{-1}$ for HO$_2$ (of which 1.10±0.79 ppbv h$^{-1}$ is from formaldehyde alone) and 0.86±0.53 ppbv h$^{-1}$ for RO$_2$, respectively."

*5. Section 6, first paragraph and P9, L18-20: your measurements show a rapid morning increase of O$_3$ concentration. This increase may not only be resulted from photochemistry but also from vertical mixing. I suggest that you integrate the net O$_3$ production rate in Figure 13 to get a diurnal profile of O$_3$ based on box model simulations and compare it with the observed diurnal profile shown in Figure 3.*

**Response:** thanks for the helpful suggestion. Indeed, the observed morning increase of O$_3$ concentrations can be due to both photochemistry and vertical mixing. We have examined the detailed budget of O$_3$ changes. The observed rate of change in O$_3$ concentrations at the study site (R$_{meas}$) was the result of chemistry (R$_{chem}$; including production and destruction), deposition (R$_{deps}$), and transport (R$_{trans}$; including horizontal and vertical transport). R$_{chem}$ and R$_{deps}$ were calculated by the observation-constrained chemical box model, and R$_{meas}$ can be derived from the observed diurnal profiles of O$_3$ concentrations. The calculated results for the nine ozone episodes are shown below. During four episodes, downward intrusion of O$_3$-laden residual layer air was clearly illustrated, as indicated by the large positive R$_{trans}$ value in the morning. However, the downward mixing mainly occurred during the early morning period (i.e., 5:00-7:00 LT), whilst the observed O$_3$ increase in the mid- and late morning should be owing to the photochemical production. For clarity, the following discussions have been added in the revised manuscript.

[Figure]

**Figure R1.** The observed rate of change in $O_3$ concentrations ($R_{meas}$) and the contributions from photochemistry ($R_{chem}$), transport ($R_{trans}$), and deposition (Rdeps) in the YelRD during the nine selected $O_3$ episodes.

"A noteworthy result is the fast accumulation of $O_3$ during the morning period. For example, the average increases in $O_3$ concentrations in the morning (06:00–12:00 LT) were 49.2 ppbv and 30.2 ppbv in summer and winter-spring, respectively. The early morning (i.e., 05:00-07:00 LT) $O_3$ increase may be attributed to the downward intrusion of $O_3$-laden residual layer air (see Fig. S1), while the rapid $O_3$ increase throughout the morning period suggests the strong in-situ photochemical formation in this VOC-rich area."

*6. Some of the box model results can be compared with those from Chinese megacities reported by Tan et al. (2019). [Tan et al., Daytime atmospheric oxidation capacity in four Chinese megacities during the photochemically polluted season: a case study based on box model simulation, Atmos. Chem. Phys., 19, 3493–3513, 2019.]*

**Response:** the findings of Tan et al. (2019) have been compared with our box model results in the revised manuscript. The revisions are as follows.

"In comparison, a recent study illustrated the importance of HONO and formaldehyde

photolysis in four polluted Chinese megacities (Beijing, Shanghai, Guangzhou and Chongqing), which accounted for ~50% of the total primary $RO_x$ source (Tan et al., 2019)."

"This is quite different from those derived from the polluted urban areas, where the $RO_x+NO_x$ reactions generally dominate the radical termination processes (Tan et al., 2019)."

Tan, Z., Lu, K., Jiang, M., Su, R., Wang, H., Lou, S., Fu, Q., Zhai, C., Tan, Q., Yue, D., Chen, D., Wang, Z., Xie, S., Zeng, L., and Zhang, Y.: Daytime atmospheric oxidation capacity in four Chinese megacities during the photochemically polluted season: a case study based on box model simulation, Atmos. Chem. Phys., 19, 3493–3513, 2019.

*7. P3, L15-18: when it comes to long-term trends of surface ozone in China, "Ma et al., Significant increase of surface ozone at a rural site, north of eastern China, Atmos. Chem. Phys., 16, 3969-3977, 2016" is one of the few important papers and should be cited.*

**Response:** this reference has been cited in the revised version.

*8. P4, L23-26: "In view of the regional scale, the observation site is constrained by both aged continental air masses transported from the Beijing-Tianjin-Hebei region and clean marine air from the Bohai Sea, making it an excellent platform to study the interaction between anthropogenic pollution and the natural background air in the North China Plain (NCP)". I do not think you can really sample "clean marine air from the Bohai Sea" or "natural background air" because you are so close to the oil wells.*

**Response:** the study site can receive the marine air from the Bohai Sea when the air comes from exactly the east, although southwesterly winds generally dominated in summer in the present study. The original statements have been revised as follows.

"In view of the regional scale, the observation site is constrained by both aged continental air masses transported from the Beijing-Tianjin-Hebei region and marine air from the Bohai Sea."

*9. P4, L27-P5, L2.: these sentences belong to section 2.2.*

**Response:** agree, and the original sentences have been separated as follows.

"Details of the sampling site can be found elsewhere (Zhang et al., 2019). Source samples were also collected from the nearby oil and gas wells to obtain the source profiles of VOCs from the oil field." (This sentence is still at the end of Section 2.1)

"All in-situ measurement instruments were housed in a temperature-controlled container, and the sampling inlets were mounted on top of the container with an altitude of about 5 m above the ground." (This sentence has been moved to Section 2.2)

10. P5, L14: what is "SHARP"? This abbreviation should be explained.

**Response:** we have explained and changed the sentence in the revised manuscript as follows:

"$PM_{2.5}$ mass concentrations were measured using a Synchronized Hybrid Ambient Real-time Particulate monitor (SHARP; *Thermo Scientific Model 5030*)."

11. P5, L21-30: did you use $O_3$ scrubbers when taking NMHCs samples?

**Response:** we didn't use $O_3$ scrubbers when taking NMHCs samples. The following statements have been added in the revised manuscript to state the potential uncertainty for our VOC observations.

"Note that $O_3$ scrubbers were not used ahead of the canisters during the sampling, and the sampled canisters were shipped to the UCI for analysis immediately after the individual field campaign. Some reactive VOC compounds (such as alkenes) may be decayed more or less during the time span from sampling to lab analysis. Thus, one should keep in mind that the VOC observations in this study may be subject to some uncertainty and the reactive compounds may be underestimated to some extent."

12. P7, L20-21: "Photolysis frequencies within the model were adjusted by the solar zenith angle and the measured $J(NO_2)$ (Saunders et al., 2003)." Why did you adjust photolysis frequencies within the model when you had the measured values?

**Response:** we are sorry that the original statement is misleading. In the model, the unmeasured photolysis frequencies (we only measured J for $NO_2$, HCHO, $O^1D$, HONO and $NO_3$) were calculated as a function of solar zenith angle (Saunders et al., 2003), and then were scaled with the measured $J(NO_2)$ values (based on the ratio of measured $J(NO_2)$ to calculated $J(NO_2)$). The statement has been revised as follows in the revised version.

"Unmeasured photolysis frequencies within the model were calculated as a function of the solar zenith angle (Saunders et al., 2003), and then were scaled with the measured $J(NO_2)$."

13. P9, L14-17: this statement applies only to summer.

**Response:** this statement has been revised as follows.

"VOCs generally showed higher levels during the nighttime or the early morning and lower mixing ratios during the day, with long-chain alkenes (comprising isoprene, 3-methyl-1-butene, 2-methyl-1-butene, alpha-pinene, and beta-pinene) as an exception that shows an opposite diurnal pattern in summer (Fig. 4)."

*14. P9, L26-28: can you estimate errors in the so obtained emission profiles?*

**Response:** the so obtained oilfield emission profiles may be subject to errors (or uncertainties) due to the limited size of source samples (i.e., 18). More comprehensive studies by taking much more samples are needed to better characterize the VOC emissions from the oilfield in China.

*15. Figure 11b: Since you single out HCHO photolysis, "OVOCs Photolysis" here should be changed to "Photolysis of OVOCs other than HCHO" or similar.*

**Response:** Figure 11b has been revised as suggested.

[Figure]

**Revised Figure 11.** Simulated average primary production rates of (a) OH, (b) HO$_2$, and (c) RO$_2$ during the summertime O$_3$ pollution episodes. The error bars indicate the standard deviations of the mean.

---

## Author Comment (AC2) · 25 Mar 2020

**Response to Reviewer 2:**

*General Comment: The impact of the O&NG exploration on the O$_3$ formation in both summer and winter is one of the key issues in the studies of tropospheric chemistry. Negative impact of the O&NG exploration in North America had already been presented in a series of field studies in both United States and Canada including the warnings of serious wintertime ozone non-attainment for the Basin landscape. The study of O&NG exploration on the atmospheric chemistry is missing in China and has been nicely filled up by the current paper based on winter and summer field studies in Yellow River Delta (YelRD) which is a place famous for open oil fields in China. The dataset obtained in this study are very valuable and the data analysis is systematic and scientifically sound. The paper has been clearly written. Overall, I suggest to publish the paper after the authors addressed the following comments.*

**Response:** we appreciate the reviewer for the positive comments and helpful suggestions. We have carefully considered and addressed all of these comments, and revised accordingly the original manuscript. Below we provide the original referee's comments *in black italics*, with our responses and changes in the manuscript in blue and red, respectively.

*Major Comment*
*1. It will be very valuable for the current paper to analyze the radical budget and the ozone production rates also for the winter campaign. And compare with the corresponding US studies to show from a chemical perspective why the ozone pollution is not appeared in the YelRD region.*

**Response:** thanks for the helpful suggestion. We have performed the same modelling analyses for eight winter-spring cases. Since few ozone episodes were encountered in winter and early spring in this study, the cases were selected mainly due to the availability of multiple NMHCs and carbonyl sampling data. The daily maximum hourly O$_3$ concentrations ranged from 40 to 98 ppbv during these cases. Overall, the model-simulated HO$_x$ levels, AOC, RO$_x$ production and recycling rate, and O$_3$ formation rate on the winter-spring cases (see Figures below) were much lower than those during the summertime O$_3$ episodes in the YelRD region as well as during the wintertime O$_3$ episodes in U.S. oilfield basins. We also compared the similarity and differences in the radical budget between winter-spring and summer. The following discussions have been added in the revised manuscript, with Figures R1-R6 being provided in the revised supplement.

"We also examined the atmospheric oxidation capacity, RO$_x$ radical budget, and O$_3$ formation for eight winter-spring cases, and the modelling results are documented in Figures S2-S7. Note that few O$_3$ episodes were encountered during the winter-spring campaign, and the cases were

selected mainly because of the availability of multiple NMHCs and carbonyls sampling data. The daily maximum hourly $O_3$ concentrations during these cases were in the range of 40-98 ppbv. Several aspects are noteworthy from the modelling results for winter-spring. First, the model-simulated $HO_x$ levels, AOC, $RO_x$ production and propagation rates, and $O_3$ formation rate were much lower than those determined for the summertime episodes. This is as expected due to the weaker solar radiation and less active photochemistry in winter-spring than in summer. Second, OH showed a 'normal' noontime concentration peak in winter-spring, which is different from the morning peak (~10:00 LT) found in summer (see Figs. 8 and S2). This was ascribed to the higher levels of $NO_x$ at the study site in winter-spring (Fig. 3), which were high enough to maintain the radical recycling from $HO_2$ to OH. Third, the partitioning of the primary $RO_x$ sources were generally similar between both seasons, despite the relatively lower contributions from the $O_3$-involved sources (i.e., $O_3$ photolysis and $O_3$+VOCs reactions). Photolysis of OVOCs other than formaldehyde was the dominant primary $RO_x$ source, followed by HONO and formaldehyde photolysis. Fourth, the radical termination processes were different between winter-spring and summer. The dominant radical sinks were the cross reactions between $NO_x$ and $RO_x$ in winter-spring, as a result of the relatively abundant ambient $NO_x$."

[Figure]

**Figure R1.** Model-simulated average diurnal variations of OH and $HO_2$ during the eight selected cases in February-March 2017. The shaded areas indicate the standard deviations of the mean.

[Figure]

**Figure R2.** Model-calculated average oxidation capacity of OH, O₃ and NO₃ during the eight selected cases in February-March 2017. The error bars indicate the standard deviations of the mean.

[Figure]

**Figure R3.** Model-calculated average OH reactivity ($K_{OH}$) and its breakdown to the major reactants during the eight selected cases in February-March 2017.

[Figure]

**Figure R4.** Model-simulated average primary production rates of (a) OH, (b) HO₂, and (c) RO₂ during the eight selected cases in February-March 2017. The error bars indicate the standard deviations of the mean.

[Figure]

**Figure R5.** Daytime average (6:00-18:00 LT) $RO_x$ budget during the eight selected cases in February-March 2017. The unit is ppb $h^{-1}$. The red, green and black lines indicate the production, destruction and recycling pathways of radicals, respectively.

[Figure]

**Figure R6.** Simulated average $O_3$ budget during the eight selected cases in February-March 2017.

*Specific Comment*

*1. Page 6, lines 1- 6, as described by the authors, the VOC samples were analyzed in the US lab which is far away. How did the authors make sure that the reactive compounds are not decayed away during the time span between sampling and the lab analysis?*

**Response:** the canisters were shipped to the US lab immediately after the field campaign, and were analyzed within two weeks. Indeed, some reactive compounds may be inevitably decayed more or less during the time span between sampling and lab analysis. We have stated this potential uncertainty of our VOC analysis in the revised manuscript, as follows.

"Note that $O_3$ scrubbers were not used ahead of the canisters during the sampling, and the sampled canisters were shipped to the UCI for analysis immediately after the individual field campaign. Some reactive VOC compounds (such as alkenes) may be decayed more or less during the time span from sampling to lab analysis. Thus, one should keep in mind that the VOC observations in this study may be subject to some uncertainty and the reactive compounds may be underestimated to some extent."

*2. The authors used "Atmospheric Oxidative Capacity" (page 6, line 17; page 11, line 17), "Atmospheric Oxidizing Capacity" (page 6, line 28; page 12, line 13), to denote AOC. I suggest the authors to use "Atmospheric Oxidation Capacity" for that. At least, it should be unified throughout the paper.*

**Response:** "Atmospheric Oxidation Capacity" has been used uniformly to denote "AOC" throughout the revised manuscript.

*3. Page 7, lines 9-11, I don't think the authors can refer the calculations of the ozone production rates simply to previous papers. The detailed equation needs to be given here and the uncertainty of the calculations is worth to be analyzed.*

**Response:** the ozone production rate (P(O$_3$)) was calculated as the sum of reaction rates for HO$_2$+NO and RO$_2$+NO reactions; the ozone loss rate (L(O$_3$)) was calculated as the sum of reaction rates for O$_3$ photolysis, O$_3$+OH, O$_3$+HO$_2$, O$_3$+VOCs, NO$_2$+OH, NO$_2$+RO, NO$_2$+RO$_2$ (minus the decomposition rate of organic nitrates), NO$_3$+VOCs, and loss of N$_2$O$_5$. The net O$_3$ production rate can be calculated as the difference between P(O$_3$) and L(O$_3$). Such calculated net ozone production rate actually denote the chemical production rate of O$_x$ (O$_x$=O$_3$+NO$_2$), and has been widely adopted in the previous studies. The detailed equations have been provided in the revised manuscript. The revised context is as follows.

"The O$_3$ chemical budget was also quantified by the model. O$_3$ production rate (P(O$_3$)) was calculated as the sum of reaction rates for HO$_2$+NO and RO$_2$+NO reactions (*E1*), and O$_3$ loss rate (L(O$_3$)) was computed as the sum of reaction rates for O$_3$ photolysis, O$_3$+OH, O$_3$+HO$_2$, O$_3$+VOCs, NO$_2$+OH, NO$_2$+RO$_2$ (minus the decomposition rate of organic nitrates), NO$_3$+VOCs, and loss of N$_2$O$_5$ (*E2*). The net O$_3$ production rate can be calculated as the difference between P(O$_3$) and L(O$_3$) (*E3*). Where, $k_i$ is the corresponding reaction constant."

$$P(O_3) = k_1[HO_2][NO] + \sum(k_2[RO_2][NO]) \tag{E1}$$

$$L(O_3) = k_3[O^1D)][H_2O] + k_4[O_3][OH] + k_5[O_3][HO_2] + \sum(k_{6i}[O_3][VOC_i]) + k_7[OH][NO_2] + \sum(k_{8i}[NO_2][RO_{2i}]) + 2\sum(k_{9i}[NO_3][VOC_i]) + 3k_{10}[N_2O_5] \tag{E2}$$

$$\text{net } P(O_3) = P(O_3) - L(O_3) \tag{E3}$$

*4. Figure 2, Y title, 'jO1D' the number 1 shall be superscript.*

**Response:** Figure 2 has been revised as suggested, see below.

[Figure]

**Revised Figure 2.** Time series of major trace gases, PM₂.₅, and meteorological parameters measured at the study site during February-March and June-July 2017.

*5. Figure 4, from the plot of the subgroups of the NMHCs, it is not clear to the readers how the high concentrations up to hundreds and thousands of ppbv of NMHCs is make up?*

**Response:** we are sorry that the unit used in Figure 4 is incorrect. The unit should be ppm, other than ppb. The box plot provides the 5th, 25th, 50th, 75th and 95th of the measurement data. We have corrected this mistake in the revised manuscript.

*6. Page 11, line 29 -30, why not compared your results to measurements of HOx radicals in NCP such as Tan et al., ACP, 2017. And Tan et al., ACP, 2018 needs updated (it is already published in ACP).*

**Response:** thanks for the suggestion. We have compared our model prediction with the measured results of Tan et al. (2017), both of which are generally comparable. The following discussion has been added in the revised manuscript. The reference of Tan et al. (2018) has been updated.

"Comparable noontime maxima HOₓ concentrations were observed at a rural site in the NCP region (Wangdu; Tan et al., 2017) and in some polluted urban areas, such as Tokyo and Houston (Kanaya et al., 2007; Mao et al., 2010)."

Tan, Z., Fuchs, H., Lu, K., Hofzumahaus, A., Bohn, B., Broch, S., Dong, H., Gomm, S., Häseler,

R., He, L., Holland, F., Li, X., Liu, Y., Lu, S., Rohrer, F., Shao, M., Wang, B., Wang, M., Wu, Y., Zeng, L., Zhang, Y., Wahner, A., and Zhang, Y.: Radical chemistry at a rural site (Wangdu) in the North China Plain: observation and model calculations of OH, $HO_2$ and $RO_2$ radicals, Atmos. Chem. Phys., 17, 663–690, http://doi.org/10.5194/acp-17-663-2017, 2017.

Tan, Z., Lu, K., Hofzumahaus, A., Fuchs, H., Bohn, B., Holland, F., Liu, Y., Rohrer, F., Shao, M., Sun, K. and Wu, Y.: Experimental budgets of OH, $HO_2$ and $RO_2$ radicals and implications for ozone formation in the Pearl River Delta in China 2014, Atmos. Chem. Phys., 2019, 7129–7150, https://doi.org/10.5194/acp-19-7129-2019, 2019.

*7. Page 12, line 21 – 29, there were a number of direct total OH reactivity measurements published for field studies in NCP and PRD by the PKU and FZJ groups. I suggest the authors shall compare those in addition to the comparison to US cities. To my knowledge, the kOH in this study is within the range of the available measurement in China.*

**Response:** thanks for the suggestion. These direct total OH reactivity observation studies have been compared against our model calculations in the revised version. The following discussions and references have been added.

"$K_{OH}$ in this study ($23.3\pm5.6$ s$^{-1}$) is significantly higher than those determined from some rural sites such as Hok Tsui ($9.2\pm3.7$ s$^{-1}$) (Li et al., 2018), Nashville ($11.3\pm4.8$ s$^{-1}$) (Martinez et al., 2003), and Whiteface Mountain (5.6 s$^{-1}$) (Ren et al., 2006a), and is comparable to those measured in some polluted areas like Beijing (10-30 s$^{-1}$) (Lu et al., 2013; Williams et al., 2016; Yang et al., 2017) and Guangzhou (20-50 s$^{-1}$) (Lou et al., 2010)."

Williams, J., Kessel, S. U., Nolscher, A. C., Yang, Y. D., Lee, Y., Yanez-Serrano, A. M., Wolff, S., Kesselmeier, J., Klupfel, T., Lelieveld, J., and Shao, M.: Opposite OH reactivity and ozone cycles in the Amazon rainforest and megacity Beijing: Subversion of biospheric oxidant control by anthropogenic emissions, Atmos. Environ., 125, 112–118, 2016.

Lou, S., Holland, F., Rohrer, F., Lu, K., Bohn, B., Brauers, T., Chang, C. C., Fuchs, H., Häseler, R., Kita, K., Kondo, Y., Li, X., Shao, M., Zeng, L., Wahner, A., Zhang, Y., Wang, W., and Hofzumahaus, A.: Atmospheric OH reactivities in the Pearl River Delta – China in summer 2006: measurement and model results, Atmos. Chem. Phys., 10(22), 11243–11260, 10.5194/acp-10-11243-2010, 2010.

Lu, K. D., Hofzumahaus, A., Holland, F., Bohn, B., Brauers, T., Fuchs, H., Hu, M., Häseler, R., Kita, K., Kondo, Y., Li, X., Lou, S. R., Oebel, A., Shao, M., Zeng, L. M., Wahner, A., Zhu, T.,

Zhang, Y. H., and Rohrer, F.: Missing OH source in a suburban environment near Beijing: observed and modelled OH and $HO_2$ concentrations in summer 2006, Atmos. Chem. Phys., 13(2), 1057–1080, http://doi.org/10.5194/acp-13-1057-2013, 2013.

Yang, Y., Shao, M., Keßel, S., Li, Y., Lu, K., Lu, S., Williams, J., Zhang, Y., Zeng, L., Nölscher, A. C., Wu, Y., Wang, X., and Zheng, J.: How the OH reactivity affects the ozone production efficiency: case studies in Beijing and Heshan, China, Atmos. Chem. Phys., 17, 7127–7142, 10.5194/acp-17-7127-2017, 2017.

*8. Figure 10, a lot of OVOCs is calculated in the model. The summed reactivity is much higher than that of their precursors (alkanes, alkenes, and aromatics) which is normally difficulty to achieve. The large accumulation of OVOCs in the model may be related with the lifetime of those OVOCs implemented in the model. Could the authors breakdown the speciation of the calculated OVOCs and compare with the measurements as mentioned in the part of methods. The comparison between model and measurements for OVOCs could help the authors to define the lifetime of the OVOCs in the model.*

**Response:** thanks for the suggestion. Actually, the measured $C_1$-$C_8$ carbonyl compounds have been used to constrain the box model. We calculated the $k_{OH}$ contributed from the 14 measured carbonyls, and compared it with that of the other model-simulated OVOC compounds. Overall, the measured carbonyls contributed to the majority of the $k_{OH}$ of total OVOCs, with an average fraction of 55%, while the other model-simulated OVOCs contributed to the remaining 45%. We also examined the lifetime of OVOCs implemented in the model. With a dry deposition velocity of 0.20-0.55 cm s$^{-1}$ and an assumed nocturnal mixing layer height of 300 m, the lifetime of OVOCs (only forced by dry deposition) at nighttime was estimated as 15-42 hours. In the revised manuscript, the original Figure 10 has been modified as follows, by separating the $k_{OH}$ of total OVOCs into those contributed from the measured and simulated components.

[Figure]

**Revised Figure 10.** Model-calculated average OH reactivity ($K_{OH}$) and its breakdown to the major reactants during the summertime $O_3$ pollution episodes.

*9. Figure 11, the OH production rate from $O_3$ photolysis seems to be a little large according to your $O_3$ concentrations presented in Figure 3. I assume you calculated the photolysis rate of $O_3$ ($O_3 + hv \rightarrow O^1D$), but the OH production needs a further reaction with $H_2O$ ($O^1D + H_2O \rightarrow OH$) which is competed by reaction with $N_2$ and $O_2$ ($O^1D+M\rightarrow O$), the yield of OH is often around 10% of the photolysis rate of $O_3$ depends on the $H_2O$ concentrations.*

**Response:** the OH production rate from $O_3$ photolysis was calculated from the reaction "$O^1D$ + $H_2O$ = OH + OH" (as $2*k*[O^1D]*[H_2O]$). In the present study, the modelling analyses were only conducted for nine severe $O_3$ episode days, when the maximum hourly $O_3$ concentrations exceeding 100 ppb every day (the hourly $O_3$ peak values exceeded 110 ppbv on 8 episodes and exceeded 130 ppbv on 5 episodes). The data presented in Figure 3 were campaign average and largely lower than those on the episode days. The calculated OH production rate from $O_3$ photolysis in this study also fell in the range reported from the previous studies in some other polluted areas (e.g., Liu et al., 2012; Lu et al., 2013; Xue et al., 2016; Tan et al., 2017).

Liu, Z., Wang, Y., Gu, D., Zhao, C., Huey, L.G., Stickel, R., Liao, J., Shao, M., Zhu, T., Zeng, L. and Amoroso, A.: Summertime photochemistry during CAREBeijing-2007: $RO_x$ budgets and $O_3$ formation, Atmos. Chem. Phys., 12, 7737–7752, https://doi.org/10.5194/acp-12-7737-2012, 2012.

Xue, L., Gu, R., Wang, T., Wang, X., Saunders, S., Blake, D., et al. (2016). Oxidative capacity and radical chemistry in the polluted atmosphere of Hong Kong and Pearl River Delta region: analysis of a severe photochemical smog episode, Atmos. Chem. Phys., 16(15), http://doi.org/10.5194/acp-16-9891-2016, 2016.

Lu, K. D., Hofzumahaus, A., Holland, F., Bohn, B., Brauers, T., Fuchs, H., Hu, M., Häseler, R., Kita, K., Kondo, Y., Li, X., Lou, S. R., Oebel, A., Shao, M., Zeng, L. M., Wahner, A., Zhu, T., Zhang, Y. H., and Rohrer, F.: Missing OH source in a suburban environment near Beijing: observed and modelled OH and $HO_2$ concentrations in summer 2006, Atmos. Chem. Phys., 13(2), 1057–1080, http://doi.org/10.5194/acp-13-1057-2013, 2013.

Tan, Z., Fuchs, H., Lu, K., Hofzumahaus, A., Bohn, B., Broch, S., Dong, H., Gomm, S., Häseler, R., He, L., Holland, F., Li, X., Liu, Y., Lu, S., Rohrer, F., Shao, M., Wang, B., Wang, M., Wu, Y., Zeng, L., Zhang, Y., Wahner, A., and Zhang, Y.: Radical chemistry at a rural site (Wangdu) in the North China Plain: observation and model calculations of OH, $HO_2$ and $RO_2$ radicals, Atmos. Chem. Phys., 17, 663–690, http://doi.org/10.5194/acp-17-663-2017, 2017.

*10. Figure 12, daytime average, please specify the exact time span of the hours*

**Response:** the time window is 06:00-18:00 local time. We have specified the exact time span

in the revised figure caption.

*11. Figure 13, the O₃ loss rate reached up to 20 ppb/h, what are the major reactions for that?*

**Response:** we are sorry that the original Figure 13 was wrong. We have corrected the calculation of the $O_x$ chemical budget, and the revised figure is shown below. Figure 13 has been modified in the revised manuscript.

[Figure]

**Revised Figure 13.** Model-simulated average chemical budgets of $O_x$ during the selected $O_3$ episodes.